# Neural attentional-filter mechanisms of listening success in middle-aged and older individuals

Sarah Tune 🄳 [1,2✉], Mohsen Alavash 🄳 [1,2], Lorenz Fiedler 🄳 [1,2,3] & Jonas Obleser 🄳 [1,2✉]

Successful listening crucially depends on intact attentional filters that separate relevant from irrelevant information. Research into their neurobiological implementation has focused on two potential auditory filter strategies: the lateralization of alpha power and selective neural speech tracking. However, the functional interplay of the two neural filter strategies and their potency to index listening success in an ageing population remains unclear. Using electro-encephalography and a dual-talker task in a representative sample of listeners (N = 155; age = 39–80 years), we here demonstrate an often-missed link from single-trial behavioural outcomes back to trial-by-trial changes in neural attentional filtering. First, we observe preserved attentional–cue-driven modulation of both neural filters across chronological age and hearing levels. Second, neural filter states vary independently of one another, demonstrating complementary neurobiological solutions of spatial selective attention. Stronger neural speech tracking but not alpha lateralization boosts trial-to-trial behavioural performance. Our results highlight the translational potential of neural speech tracking as an individualized neural marker of adaptive listening behaviour.

[1] Department of Psychology, University of Lübeck, Lübeck, Germany. [2] Center for Brain, Behavior, and Metabolism, University of Lübeck, Lübeck, Germany. [3] Present address: Eriksholm Research Centre, Snekkersten, Denmark. ✉email: sarah.tune@uni-luebeck.de; jonas.obleser@uni-luebeck.de

Real-life listening is characterized by the concurrence of sound sources that compete for our attention[1]. Successful speech comprehension thus relies on the differentiation of relevant and irrelevant inputs. Here, the concept of neural attentional 'filters' serves as an important and pervasive algorithmic metaphor of how auditory attention is implemented at the neural level[2–4]. Neural attentional filters can be instantiated by different mechanistic principles and recent studies have predominantly focused on two potential but nonexclusive neural filter strategies originating from distinct research traditions:

From the visual domain stems an influential line of research that supports the role of alpha-band (~8–12 Hz) oscillatory activity in the implementation of controlled, top-down suppression of behaviourally irrelevant information[5–8]. Importantly, across modalities, it was shown that spatial-attention tasks are neurally supported by hemispheric lateralization of alpha power over occipital, parietal but also the respective sensory cortices[9–18]. This suggests that asymmetric alpha modulation could act as a filter mechanism by modulating sensory gain already in the early processing stages.

In addition, a prominent line of research focuses on the role of low-frequency (1–8 Hz) neural activity in auditory and, broadly speaking, perisylvian cortex in the selective representation of speech input ('neural speech tracking'). Slow cortical dynamics temporally align with (or 'track') auditory input signals to prioritize the neural representation of behaviourally relevant sensory information[19–22] (see also refs. [23,24] for the neural tracking of contextual semantic information). In human speech comprehension, a key finding is the preferential neural tracking of attended compared to ignored speech in superior temporal brain areas close to the auditory cortex[25–29].

However, with few exceptions[9], these two proposed neural auditory filter strategies have been studied independently of one another (but see refs. [30,31] for recent results on visual attention). Also, they have often been studied using tasks that are difficult to relate to natural, conversation-related listening situations[32,33].

We thus lack understanding of whether or how modulations in lateralized alpha power and the neural tracking of attended versus ignored speech in the wider auditory cortex interact in the service of successful listening behaviour. Moreover, few studies using more real-life listening and speech-tracking measures were able to explicitly address the functional relevance of the discussed neural filter strategies, that is, their potency to explain behavioural listening success[27,28].

As part of an ongoing large-scale project on the neural and cognitive mechanisms supporting adaptive listening behaviour in healthy ageing, this study aims at closing these gaps by leveraging the statistical power and representativeness of our large, age-varying participant sample. We use a dichotic listening paradigm to enable a synergistic look at concurrent single-trial changes in lateralized alpha power and neural speech tracking.

More specifically, our linguistic variant of a classic Posner paradigm[34] emulates a challenging dual-talker listening situation, in which speech comprehension is supported by two different listening cues[35,36]. Of particular interest for the present scientific endeavour is the spatial-attention cue that guides auditory attention in space. We additionally manipulated the semantic predictability of upcoming speech via a semantic category cue. While the effects of the semantic cue are of secondary importance for the present research questions, its manipulation still allows insights into whether semantic predictability modulates the engagement of neural attentional filter mechanisms, and how it affects listening success in a large cohort of middle-aged and older adults. Previous research has shown that the sensory analysis of speech and, to a lesser degree, the modulation of alpha power are influenced by the availability of higher-order linguistic information[37–42].

Varying from trial to trial, both cues were presented either in an informative or uninformative version. This manipulation allowed us to understand how concurrent changes in the neural dynamics of selective attention and the resulting listening behaviour are connected.

We focus on four main research questions (see Fig. 1). Note that in addressing these, we model additionally known influences on listening success: age, hearing loss, as well as hemispheric asymmetries in speech processing due to the well-known right-ear advantage[43,44].

First, informative listening cues should increase listening success: these cues allow the listener to deploy auditory selective attention (compared to divided attention), and to generate more specific (compared to only general) semantic predictions, respectively.

Second, we asked how the different cue–cue combinations would modulate the two key neurobiological measures of selective attention—alpha lateralization and neural speech tracking. We aimed to replicate previous findings of increased alpha lateralization and a preferential tracking of the target compared to the distractor speech signal under selective (compared to divided) spatial attention. At the same time, we capitalized on our age-varying sample to quantify the hitherto contested dependence of these neurobiological filters on participants' chronological age and hearing loss[14,45–47].

Third, an important and often neglected research question pertains to a direct, trial-by-trial relationship of these two candidate neural measures: Do changes in alpha lateralization impact the degree to which attended and ignored speech signals are neurally tracked by low-frequency cortical responses?

Our final research question is arguably the most relevant one for all translational aspects of auditory attention; it has thus far only been answered indirectly when deeming these neurobiological filter mechanisms 'attentional': to what extent do alpha lateralization and neural speech tracking allow us to explain the behavioural outcome—that is, listening success—at the individual level and in a single trial?

Here, we show how an attentional cue modulates neural speech tracking and alpha lateralization independently of age and hearing levels. We demonstrate the co-existence of largely independent neural filters that pose complementary neurobiological implementations of selective attention. Stronger neural speech tracking

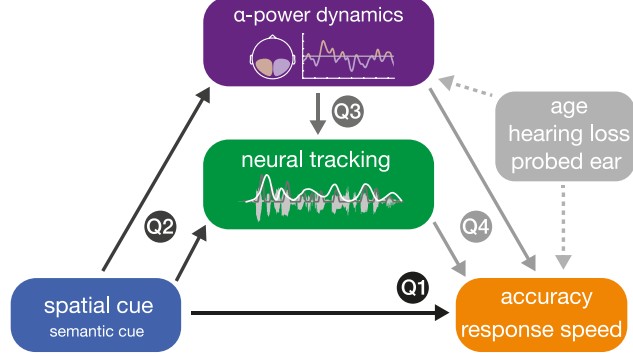

**Fig. 1 Schematic illustration of addressed research questions.** The dichotic listening task manipulated the attentional focus and semantic predictability of upcoming input using two separate visual cues. We investigated whether informative cues would enhance behavioural performance (Q1). In line with (Q2), we also examined the degree to which a spatial (and semantic) cue modulated the two auditory neural measures of interest: neural speech tracking and lateralization of auditory alpha power. Finally, we assessed (Q3) the co-variation of neural measures, and (Q4) their potency in explaining behavioural performance. Furthermore, we investigated the impact of age, hearing loss, and probed ear on listening success and its underlying neural strategies.

but not alpha lateralization increases trial-to-trial listening performance. This emphasizes the potential of neural speech tracking as a diagnostic neural measure of an individual's listening success.

## Results

We recorded and source-localized electroencephalography (EEG) signals in an age-varying sample of healthy middle-aged and older adults ($N = 155$; age = 39–80 years, see Supplementary Fig. 1) who performed a challenging dichotic listening task. In this linguistic variant of a classic Posner paradigm[35], participants listened to two concurrent five-word sentences spoken by the same female talker and were asked to identify the final word in one of the two sentences. Sentence pairs were temporally aligned to the onset of these task-relevant final words which led to slightly asynchronous sentence onsets.

Importantly, sentence presentation was preceded by two visual cues. First, a spatial-attention cue encouraged the use of either selective or divided attention by providing informative or uninformative instructions about the to-be-attended, and thus later-probed, ear. The second cue indicated the semantic category that applied to both final target words. The provided category could represent a general or specific level, thus allowing for more or less precise prediction of the upcoming speech signal (Fig. 2a, b). While this listening task does not tap into the most naturalistic forms of speech comprehension, it still approximates a dual-talker listening situation to probe the neural underpinnings of successful selective listening[35].

Using generalized linear mixed-effects models on single-trial data, we focus on two key neurobiological instantiations of auditory attention: the lateralization of 8–12 Hz alpha power, emerging from auditory as well as parietal cortex, and the differential neural tracking of attended versus ignored speech by slow (1–8 Hz) auditory cortical responses. We investigate how spatial cues, age and hearing status modulate behaviour and neural filters, whether neural filters operate independently, and to which extent they influence selective listening success.

### Informative spatial cues improve listening success.
For behavioural performance, we tested the impact of informative versus uninformative cues on listening success. Overall, participants achieved a mean accuracy of 87.8% ± SD 9.1% with a mean reaction time of 1742 ms ± SD 525 ms; as response speed: 0.62 s$^{-1}$ ± SD 0.17 s$^{-1}$.

As expected, behaviour depended on the different combinations of listening cues (Fig. 2c, d). Informative compared to uninformative spatial-attention cues yielded a strong behavioural benefit. In selective-attention trials, participants responded more accurately and faster (accuracy: generalized linear mixed-effects model (GLMM); odds ratio (OR) = 3.5, std. error (SE) = 0.12, $P < 0.001$; response speed: linear mixed-effects model (LMM); $\beta$ = 0.57, SE = 0.04, $P < 0.001$; see Supplementary Tables 1 and 2). That is, when cued to one of the two sides, participants responded on average 261 ms faster and their probability of giving a correct answer increased by 6%.

Also, participants responded generally faster in trials in which they were given a specific, more informative semantic cue (LMM; $\beta$ = 0.2, SE = 0.03, $P < 0.001$), most likely reflecting a semantic priming effect that led to faster word recognition. Contrary to our expectations, a more informative semantic cue did not lead to more accurate responses (GLMM; OR = 1.1, SE = 0.11, $P = 0.69$).

As in a previous fMRI implementation of this task[35], we did not find evidence for any interactive effects of the two listening cues on either accuracy (GLMM; OR = 1.3, SE = 0.21, $P = 0.36$) or response speed (LMM; $\beta$ = 0.09, SE = 0.06, $P = 0.31$). Moreover, the breakdown of error trials revealed a significantly higher

proportion of spatial stream confusions (6% ± SD 8.3%) compared to random errors (3% ± SD 3.4%; paired $t$ test on logit-transformed proportions: $t_{155}$ = 6.53, $P < 0.001$; see Supplementary Fig. 2). The increased rate of spatial stream confusions (i.e., responses in which the last word of the to-be-ignored sentence was chosen) attests to the distracting nature of dichotic sentence presentation and thus heightened task difficulty.

### Spatial attention modulates both alpha lateralization and neural speech tracking in the auditory cortex.
In line with our second research question, following source projection of EEG data, we probed whether the presence of an informative spatial-attention cue would lead to reliable modulation of both alpha power and neural speech tracking within an a priori defined auditory region of interest (ROI; see Supplementary Fig. 3 and Supplementary Methods for details).

For alpha power, we expected attention-induced lateralization due to a decrease in power contralateral and a concomitant increase in power ipsilateral to the focus of attention. For neural speech tracking, we expected stronger neural tracking of attended compared to ignored speech under selective attention but no such systematic difference in the neural tracking of probed and unprobed sentences in divided-attention trials. Accordingly, our analyses of alpha power and neural speech tracking focused on attentional modulation index measures that contrast the relative strength of neural responses to target versus distractor stimuli. In line with previous results, we expected alpha lateralization to be present throughout the auditory sentence presentation but to potentially increase around the task-relevant final word[12,14,48].

We compared alpha-power changes ipsi- and contralateral to the probed ear to derive a temporally resolved single-trial measure of alpha-power lateralization [alpha lateralization index (ALI) = ($\alpha$-power$_{ipsi}$ − $\alpha$-power$_{contra}$)/($\alpha$-power$_{ipsi}$ + $\alpha$-power$_{contra}$)][15].

As shown in Fig. 3a, an informative spatial cue—that is, the instruction to pay attention to a given side—elicited pronounced lateralization of 8–12 Hz alpha power within the auditory ROI. Lateralization of alpha power was evident following the spatial cue itself and during dichotic sentence presentation with its strongest peak around final word presentation.

As expected, the statistical analysis of alpha lateralization during sentence presentation (time window: 3.5–6.5 s; see control analysis section below for results on the final-word period) revealed a significant modulation by the attention that was additionally influenced by the probed ear (LMM; spatial cue x probed ear: $\beta$ = 0.13, SE = 0.02, $P < 0.001$; Fig. 3b). Follow-up analysis showed a significant difference in alpha lateralization between selective- and divided-attention trials when the right ear but not when the left was probed (LMM, right-ear probed: $\beta$ = 0.12, SE = 0.01, $P < 0.001$; LMM, left-ear probed: $\beta$ = 0.016, SE = 0.013, $P = 0.55$; see Supplementary Table 3). This pattern suggests that when given an uninformative spatial cue, participants presumably payed overall more attention to the left-ear stimulus leading to an increase in alpha lateralization for probed-left compared to probed-right trials.

Notably, we did not find any evidence for modulation by the semantic cue nor any joint influence of the spatial and semantic cue on alpha lateralization during sentence presentation (LMM; semantic cue main effect: $\beta$ = −0.01, SE = 0.01, $P = 0.53$, spatial × semantic cue: $\beta$ = −0.02, SE = 0.02, $P = 0.53$).

Not least, the extent of overall as well as attention-specific alpha lateralization was unaffected by participants' chronological age and hearing loss ($P$ values > 0.27 for main effects of age, PTA, and their respective interactions with the spatial-attention cue; see also Supplementary Table 6 for a corresponding analysis of alpha power during the interval of the final word).

In close correspondence to the alpha-power analysis, we investigated whether changes in attention or semantic

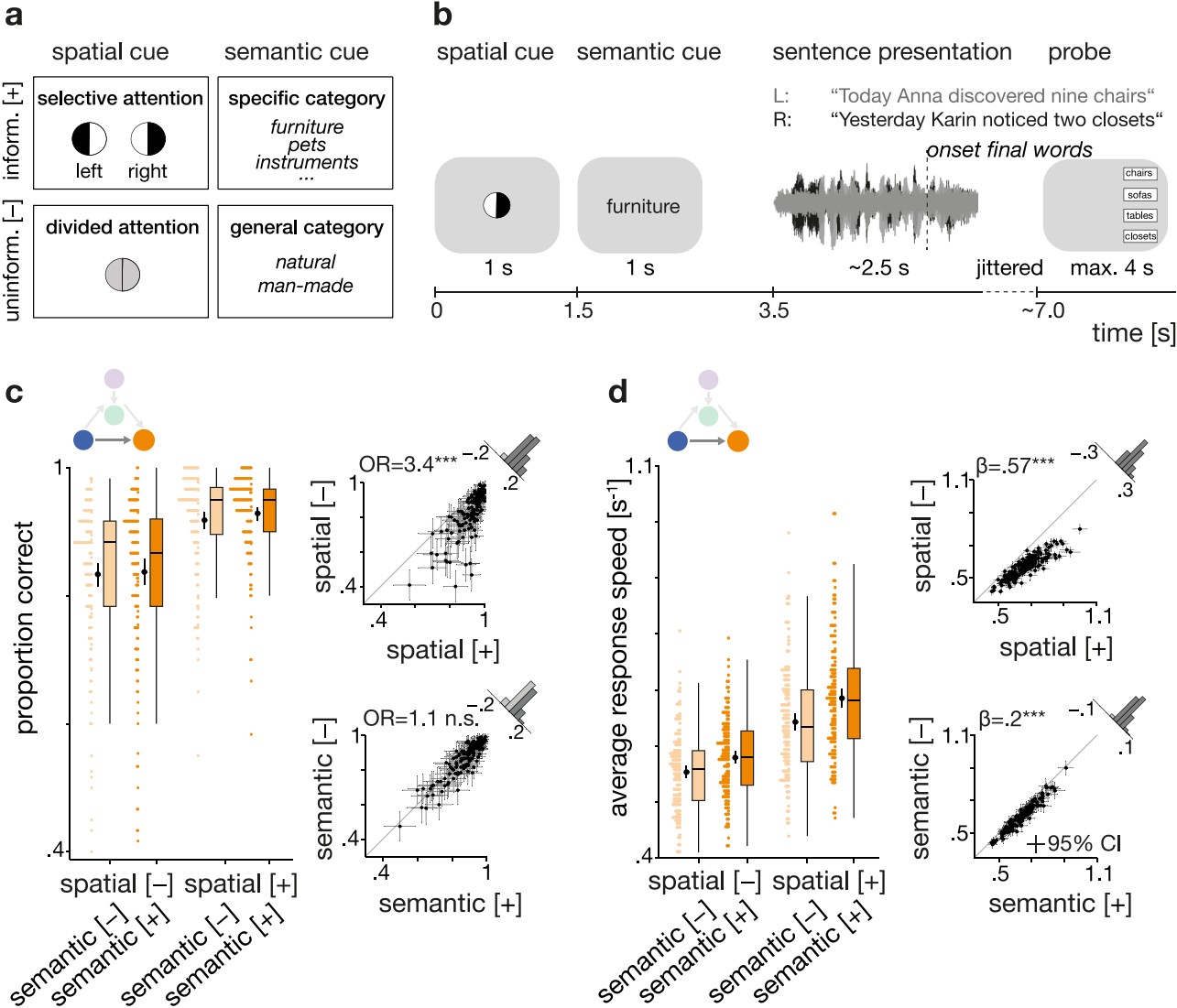

**Fig. 2 Experimental design and behavioural benefit from informative cues. a** Visualization of used 2 × 2 design[35]. Levels of spatial and semantic cues differed on a trial-by-trial basis. Note that the effects of the semantic cue were of secondary importance to the current analyses. Top row shows the informative [+] cue levels, bottom row the uninformative [−] cue levels. **b** Schematic representation of the trial structure. Successive display of the two visual cues precedes the dichotic presentation of two sentences spoken by the same female talker. After sentence presentation, participants had to select the final word from four alternative words. **c** Left: accuracy per cue–cue combination. Coloured dots are individual (N = 155 participants) trial averages, black dots and vertical lines show group means with bootstrapped 95% confidence intervals (CI). Right: Individual cue benefits displayed separately for the two cues (top: spatial cue, bottom: semantic cue). Black dots indicate individual (N = 155) mean accuracy with bootstrapped 95 % CI. Histograms show the distribution of the difference in accuracy for informative vs. uninformative levels. OR: odds ratio parameter estimate from generalized linear mixed-effects models; two-sided Wald test (FDR-corrected); spatial cue: $P = 1.36 \times 10^{-24}$; semantic cue: $P = 0.68$. **d** Left: Response speed per cue–cue combination. Coloured dots show individual (N = 155 participants) mean speed, black dots and vertical lines show group means with bootstrapped 95% CI. Right: Individual cue benefits displayed separately for the two cues (top: spatial cue, bottom: semantic cue). Black dots indicate individual (N = 155) mean speed with bootstrapped 95% CI. Box plots in (**c**) and (**d**) show median centre line, 25th to 75th percentile hinges, whiskers indicate minimum and maximum within 1.5 × interquartile range. β: slope parameter estimate from general linear mixed-effects models; two-sided Wald test (FDR-corrected); spatial cue: $P = 4.49 \times 10^{-48}$; semantic cue: $P = 2.49 \times 10^{-9}$. Source data are provided as a Source Data file.

predictability would modulate the neural tracking of attended versus ignored speech. We used linear backward ('decoding') models to reconstruct the onset envelopes of the to-be-attended and ignored sentences (for simplicity hereafter referred to as attended and ignored) from neural activity in the auditory ROI. Reconstruction models were trained on selective-attention trials only but then utilized to reconstruct attended (probed) and ignored (unprobed) envelopes for both attention conditions (see 'Methods', Fig. 4a and Supplementary Fig. 4 for details).

In line with previous studies[26,28,49], the forward-transformed temporal response functions (TRFs) show increased encoding of attended compared to ignored speech in the time window covering the $N1_{TRF}$ and $P2_{TRF}$ component (see Fig. 4b, left panel). Here, however, this was observed particularly for right-ear inputs processed in the left auditory ROI.

Further attesting to the validity of our reconstruction models, reconstructed attended envelopes were overall more similar to the envelope of the to-be-attended sentence than to that of the

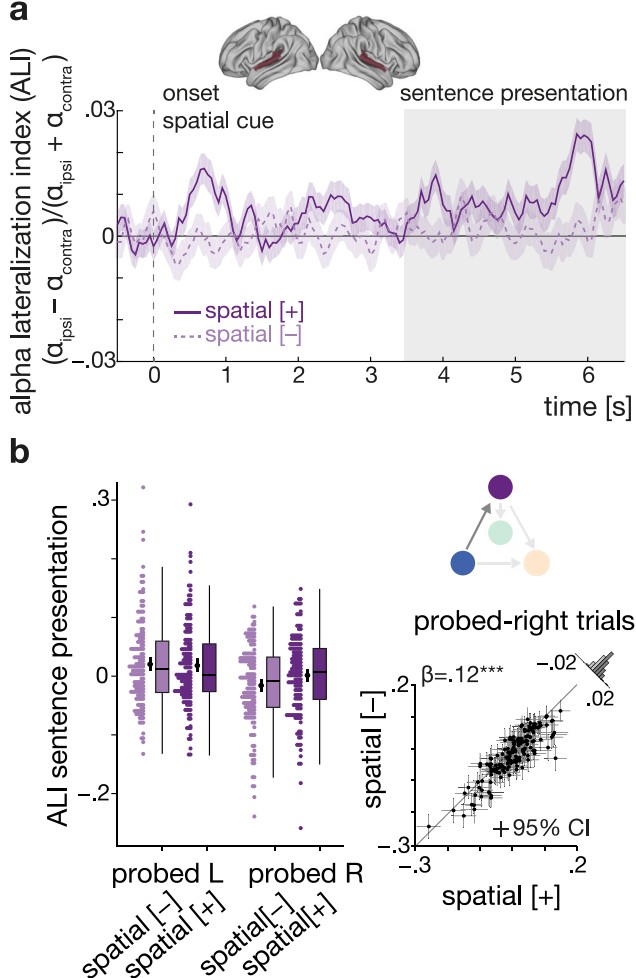

**Fig. 3 Informative spatial cue elicits increased alpha-power lateralization before and during speech presentation. a** Grand-average ($N = 155$ participants) whole-trial attentional modulation of 8–12 Hz auditory alpha power. Purple traces show the grand-average alpha lateralization index (ALI) for the informative (solid dark purple line) and uninformative spatial cue (dashed light purple line), each collapsed across semantic cue levels. Error bands indicate ±1 SEM. Positive values indicate relatively higher alpha power in the hemisphere ipsilateral to the attended/probed sentence compared to the contralateral hemisphere. The shaded grey area shows the time window of sentence presentation. Brain models show the auditory region of interest (red). **b** ALI during sentence presentation (3.5–6.5 s) shown separately per spatial-cue condition and probed ear (left plot) for $N = 155$ participants. Coloured dots show trial-averaged individual results, black dots and error bars indicate the grand-average and bootstrapped 95% confidence intervals. Box plots show median centre line, 25th to 75th percentile hinges; whiskers show minimum and maximum within 1.5 × interquartile range. For probed-right trials, there was a significant difference in ALI between selective- and divided-attention trials (right plot). Black dots represent individual mean ALI values with bootstrapped 95% CI error bars. Histogram shows the distribution of differences in ALI for informative vs. uninformative spatial-cue levels. β: slope parameter estimate from the corresponding general linear mixed-effects model; two-sided Wald test (FDR-corrected, ***$P = 2.65 \times 10^{-17}$). Source data are provided as a Source Data file.

to-be-ignored sentence, and vice versa for the reconstructed ignored envelopes (see Fig. 4b, right panel).

As shown in Fig. 4c, the differential neural tracking of attended and ignored envelopes (probed and unprobed envelopes under divided attention) was modulated by attention. Following an

informative spatial cue, the neural tracking index becomes increasingly positive during the second half of sentence presentation with its highest peaks around final-word onset.

The statistical analysis of single-trial index values averaged for the time interval of final-word presentation confirmed this pattern: the difference in the neural tracking of the attended and ignored sentence was generally more pronounced under selective compared to divided attention (see control analysis section below for results on the entire sentence presentation). However, this effect was also modulated by differences in sentence onset: the difference in neural speech tracking between the two attention conditions was reduced when the attended/probed sentence started ahead of the distractor sentence. This effect was driven by an increase in differential neural speech tracking for divided attention in such trials: in absence of an informative spatial cue, participants' attention was captured by the sentence with the earlier onset. Consequently, we observed overall more positive index values when the earlier sentence was probed compared to when it was not probed (LMM, earlier onset × spatial cue: $\beta = -0.05$, SE = 0.02, $P = 0.049$, see Fig. 4d and Supplementary Table 4 for full model details).

We also found a neural correlate of the known right-ear advantage for verbal materials, that is, an overall stronger tracking of left-ear inputs. This effect was independent of spatial-attention cueing (LMM; probed ear main effect: $\beta = -0.03$, SE = 0.01, $P = 0.023$; spatial cue × probed ear: $\beta = 0.02$, SE = 0.02, $P = 0.54$). As for alpha power, we did not observe any modulation of neural tracking by the semantic cue, nor any joint influence of the spatial and semantic cue (LMM; semantic cue main effect: $\beta = 0.01$, SE = 0.01, $P = 0.53$, interaction spatial × semantic cue: $\beta = -0.02$, SE = 0.02, $P = 0.53$).

Again, participants' age and hearing status did not prove significant predictors of neural speech tracking ($P$ values > 0.54 for main effects of age, PTA, and their respective interactions with the spatial-attention cue, see also Supplementary Table 7 for a corresponding analysis of neural tracking during the entire sentence presentation).

**Trial-to-trial neural speech tracking is independent of synchronous alpha lateralization.** Our third major goal was to investigate whether neural speech tracking and the modulation of alpha power reflects two dependent neural mechanisms of auditory attention at all. We asked whether neural speech tracking could be explained by auditory alpha lateralization either at the state level (i.e., within an individual from trial to trial) or at the trait level (i.e., between individual mean levels; see 'Statistical analysis' for details). If modulations of alpha power over auditory cortices indeed act as a neural filter mechanism to selectively gate processing during early stages of sensory analysis, then heightened levels of alpha lateralization should lead to a more differentiated neural tracking of the attended vs. ignored speech and thus a more positive neural tracking index (cf. Fig. 5a).

However, in the analysis of the task-relevant final-word period, we did not find evidence for an effect of alpha lateralization on neural speech tracking at neither the state nor the trait level (Fig. 5b, LMM; ALI within-subject effect: $\beta = -0.008$, SE = 0.005, $P = 0.35$; ALI between-subject effect: $\beta = -0.0007$, SE = 0.007, $P = 0.98$; see Supplementary Table 5). This notable absence of an alpha lateralization–neural speech tracking relationship held irrespective of spatial-attention condition or probed ear (all $P$ values > 0.35).

To complement our fine-grained single-trial level investigation into the brain–brain relationship with a coarser, yet time-resolved analysis, we related the temporal dynamics of both neural measures in an exploratory between-subjects cross-correlation

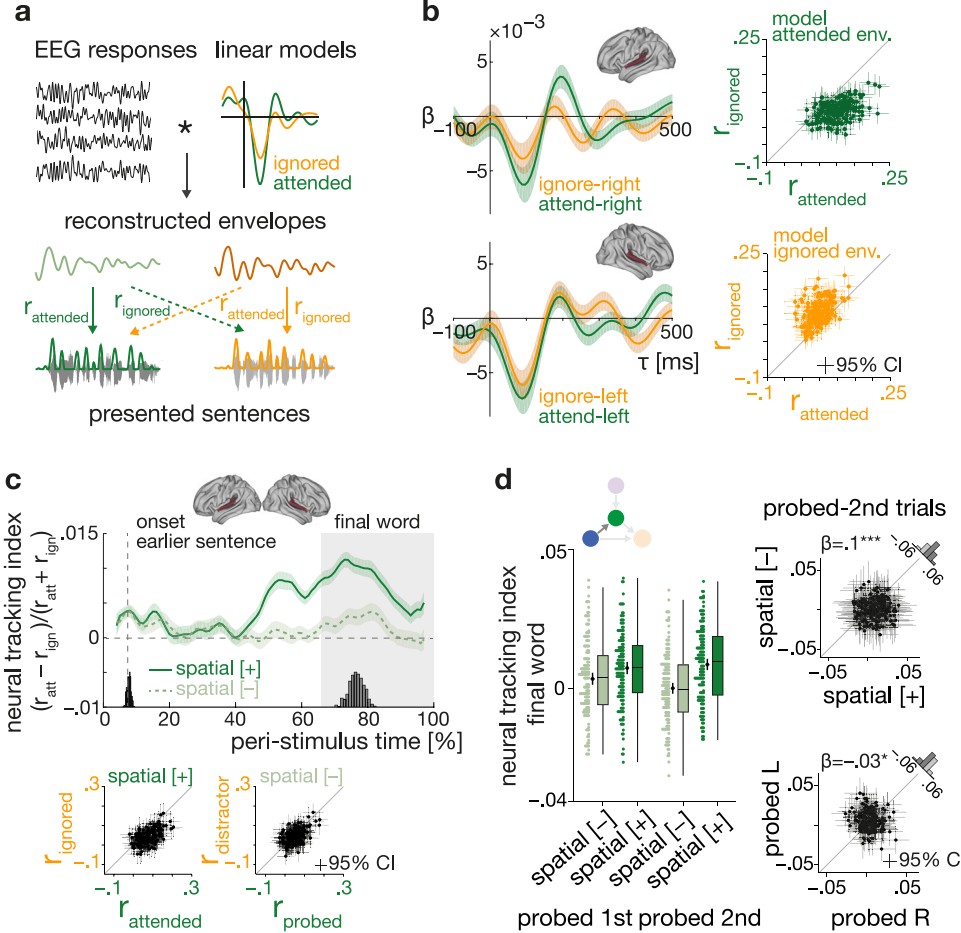

**Fig. 4 Neural speech tracking of attended and ignored sentences. a** Schematic representation of linear backward model approach. Linear backward models estimated on selective-attention trials. Onset envelopes are reconstructed via convolution of auditory EEG responses with estimated backward models and compared actual envelopes to assess neural tracking strength and decoding accuracy (see Supplementary Methods). **b** Left: grand-average (N = 155 participants, 95% confidence interval (CI) error bands) forward-transformed temporal response functions (TRFs) for attended (green) and ignored (yellow) speech in the left and right auditory ROI. Right: single-subject (N = 155 participants; 95% CI error bars) mean Pearson correlation of reconstructed and presented envelopes shown separately for attended (top, green) and ignored speech (bottom, yellow). **c** Top: grand-average (N = 155 participants) peri-stimulus time course of neural tracking index shown separately for selective (solid dark green curve) and divided attention (dashed light-green curve) ±1 SEM error band. Histograms show sentence and final-word onsets. The shaded area indicates the final-word presentation interval used for statistical analysis. Bottom: Single-subject (N = 155 participants) mean attended and ignored neural speech tracking during final-word presentation for selective and divided attention, respectively. **d** Left: neural tracking index shows per spatial-attention condition and for trials in which cued/probed sentences started ahead of ('probed first') or after the distractor ('probed second'). Coloured dots represent the single-subject average (N = 155 participants), black dots and error bars indicate grand-average and bootstrapped 95% CI. Box plots show median centre line, 25th to 75th percentile hinges, whiskers indicate minimum and maximum within 1.5 × interquartile range. Right: significant difference in neural tracking between selective- and divided-attention trials in probed second trials (top plot), and stronger neural tracking in probed-left trials. Black dots represent the individual mean neural tracking index with bootstrapped 95% CI error bars for N = 155 participants. Histogram shows the distribution of differences in neural tracking in informative vs. uninformative spatial-cue trials, and probed-left vs. probed-right trials, respectively. β: slope parameter estimate from the corresponding general linear mixed-effects model; \*\*\*P = 1.22 × 10^−9, \*P = 0.0233 (two-sided Wald test, FDR-corrected). Source data are provided as a Source Data file.

analysis. As shown in Fig. 5c, under selective attention, neural speech tracking and alpha lateralization follow different temporal trajectories with neural speech tracking peaking earlier than alpha lateralization around final-word presentation. The average cross-correlation of the two neural time courses during sentence presentation confirms a systematic temporal delay with fluctuations in neural speech tracking leading those in alpha power by about 520 ms (see Fig. 5d).

**Neural speech tracking but not alpha lateralization explains listening behaviour.** Having established the functional independence of alpha lateralization and neural speech tracking at the

single-trial level, the final piece of our investigation was to probe their relative functional importance for behavioural outcomes.

Using the same (generalized) linear mixed-effects models as in testing our first research question (Q1 in Fig. 1), we investigated whether changes in task performance could be explained by the independent (i.e., as main effects) or joint influence (i.e., as an interaction) of neural measures. Again, we modelled the influence of the two neural filter strategies on behaviour at the state and trait level[50].

For response accuracy, our most important indicator of listening success, we observed an effect of trial-by-trial variation in neural speech tracking both during the presentation of the final word and across the entire sentence: participants had a higher

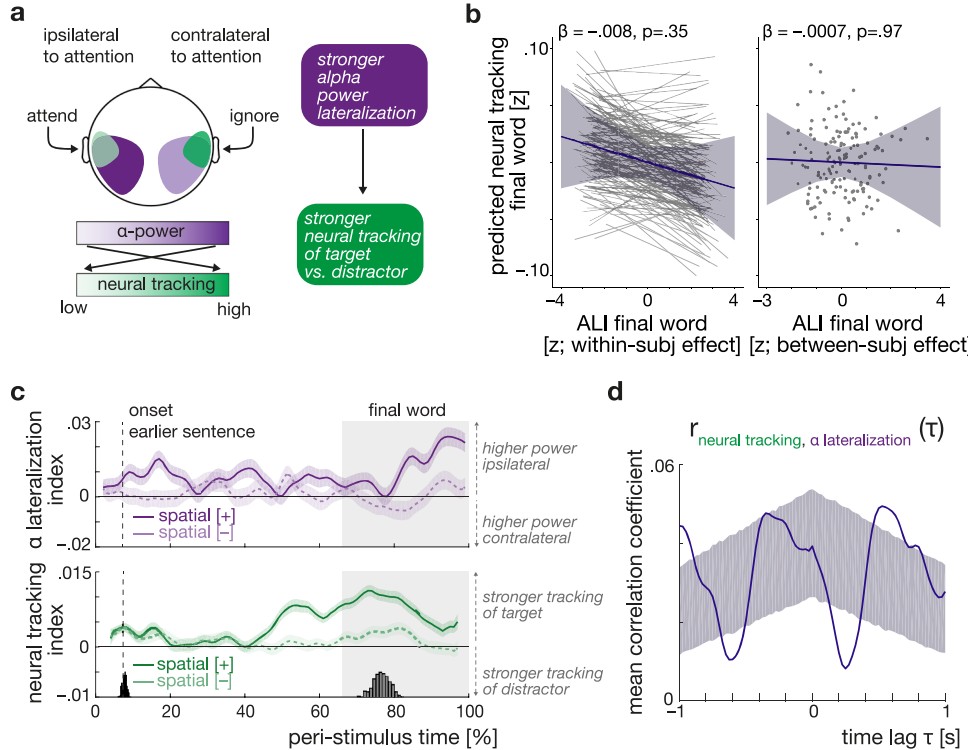

**Fig. 5 Relationship of alpha lateralization and neural speech tracking. a** Hypothesized relationship of alpha power and neural speech tracking within the auditory region of interest. Changes in alpha lateralization are assumed to drive changes in neural tracking. Schematic representation for an attend-left trial. **b** Independence of neural speech tracking and alpha lateralization during final-word presentation as shown by the predictions from the same linear mixed-effect model. Plots show the predicted, non-significant effect of within- and between-subject variations in alpha lateralization on selective neural tracking, respectively. Blue lines indicate the respective predicted fixed effects with 95% confidence interval, grey thin lines in the left plot show $N = 155$ subject-specific random slopes (included for illustrative purposes only), and grey dots show average predictions per subject. β: slope parameter estimate from the corresponding general linear mixed-effects model, two-sided Wald test (FDR-corrected). **c** Grand-average time courses of alpha lateralization and neural speech tracking during sentence presentation mapped to the same peri-stimulus time axis. Shown separately for selective attention (darker, solid curves) and divided-attention trials (lighter, dashed curves). Error bands reflect ±1 SEM. Note how the peak in neural speech tracking under selective attention precedes the peak in alpha lateralization. **d** Mean normalized cross-correlation of trial-averaged neural speech tracking and alpha lateralization time courses. The upper and lower bound of the shaded areas reflect the 97.5th and 2.5th percentile of surrogate data derived from 5000 independently permuted time courses of alpha power and neural speech tracking. Source data are provided as a Source Data file.

chance of responding correctly in trials in which they neurally tracked the cued/probed sentence more strongly than the distractor sentence (see Fig. 6a, left panel). For changes in neural speech tracking extracted from the entire sentence presentation, this effect occurred independently of other modelled influences (GLMM; main effect neural tracking (within-subject effect): OR = 1.06, SE = 0.02, $P = 0.03$; see Supplementary Table 12) while it was generally less pronounced and additionally modulated by the probed ear for the period of the task-relevant final word (GLMM; probed ear × neural tracking (within-subject effect): OR = 1.1, SE = 0.04, $P = 0.03$; see Supplementary Table 1).

The data held no evidence for any direct effects of trial-to-trial or participant-to-participant variation in alpha lateralization during a sentence or final-word presentation on accuracy (all $P$ values > 0.18; see Supplementary Tables 1 and 12). We also did not find evidence for any joint effects of alpha power and neural speech tracking extracted from either of the two time windows (all $P$ values > 0.33). Importantly, the absence of an effect did not hinge on differences in neural measures across spatial-cue, or probed-ear levels as relevant interactions of neural measures with these predictors were included in the model (all $P$ values > 0.55).

The observed effects of neural filters on response speed depended on the analysed time window: while participants with relatively higher average levels of neural speech tracking during sentence presentation responded overall faster (LMM, neural tracking (between-subject effect): $\beta = 0.08$, SE = 0.03, $P = 0.01$; see Fig. 6a, right panel and Supplementary Table 13), we found a combined effect of neural dynamics during final-word presentation. Under selective but not divided attention, response speed depended on a combination of trial-to-trial variation in both alpha lateralization and neural speech tracking (LMM; spatial cue × ALI (within-subject effect) × neural tracking index (within-subject effect): $\beta = 0.08$, SE = 0.03, $P = 0.01$; see Supplementary Table 2). In short, responses were fastest in trials where relatively elevated levels in either neural speech tracking or alpha lateralization were paired with relatively reduced levels in the respective other neural measures thus highlighting the influence of two independent complementary filter solutions (see also Supplementary Fig. 5).

In line with the literature on listening behaviour in ageing adults[51,52], the behavioural outcome was further reliably predicted by age, hearing loss, and probed ear. We observed that participants' performance varied in line with the well-attested right-ear advantage (REA, also referred to as left-ear disadvantage) in the processing of linguistic materials[44]. More specifically, participants responded both faster and more accurately when they were probed on the last word presented to the right compared to the left ear (response speed: LMM; $\beta = 0.08$, SE =

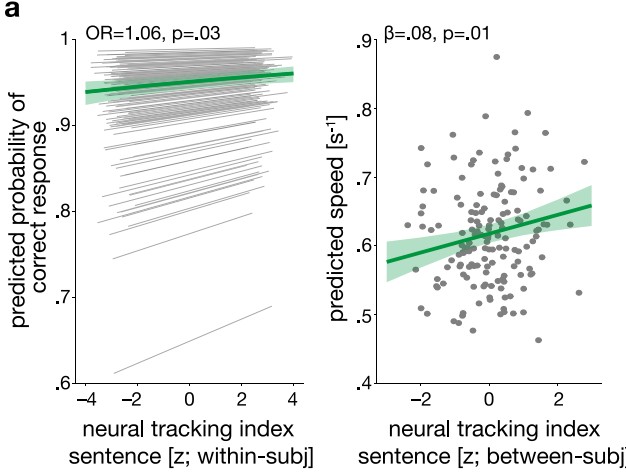

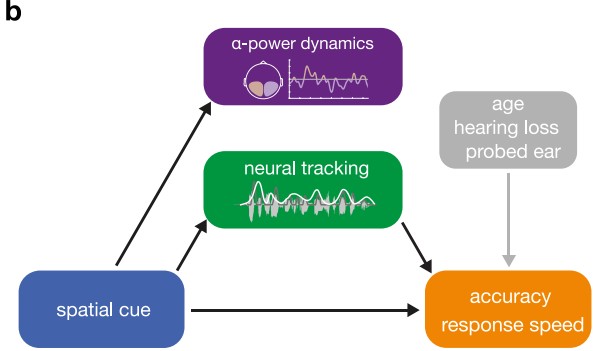

**Fig. 6 Neural speech tracking predicts listening behaviour. a** Model predictions for the effect of neural tracking on behaviour for $N = 155$ participants. Left panel shows the predicted group-level fixed effect (green line ± 95% CI) of trial-to-trial variation in neural tracking on accuracy. Grey thin lines indicate estimated subject-specific slopes. Right panel shows the predicted group-level fixed effect of neural tracking at the between-subject level on response speed (green line ± 95% CI). Grey dots indicate subject-level model predictions. OR: odds ratio, $\beta$: slope parameter estimate from the corresponding general linear mixed-effects model, two-sided Wald test (FDR-corrected). **b** Summary of results. Black arrows highlight statistically significant effects from (generalized) single-trial linear mixed-effects modelling. Grey arrow shows the effect of additionally modelled influences. Notably, changes in age and hearing loss did not modulate the fidelity of the two key neural measures. Source data are provided as a Source Data file.

0.013, $P < 0.001$; accuracy: GLMM; OR = 1.25, SE = 0.07, $P = 0.006$; see also Supplementary Fig. 6).

Increased age led to less accurate and slower responses (accuracy: GLMM; OR = 0.80, SE = 0.08, $P = 0.025$; response speed: LMM; $\beta = -0.15$, SE = 0.03, $P < 0.001$). In contrast, increased hearing loss led to less accurate (GMM; OR = 0.75, SE = 0.08, $P = 0.002$) but not slower responses (LMM; $\beta = -0.05$, SE = 0.03, $P = 0.21$, see Supplementary Tables 1–2, and Supplementary Fig. 7).

**Control analyses**. We ran additional control analyses to validate our main set of results. First, we asked whether the observed independence of alpha lateralization and neural speech tracking hinged on the precise time window and cortical site from which neural measures were extracted. One set of control models thus included alpha lateralization during spatial cue rather than sentence presentation as a predictor of neural

speech tracking during sentence and final-word presentation. Next, we related the two neural filters during the entire sentence presentation rather than only during the final word.

Second, we tested the hypothesis that neural speech tracking might be driven not primarily by alpha-power modulations emerging in auditory cortices, but rather by those generated in domain-general attention networks in the parietal cortex[53]. We, therefore, ran control models including alpha lateralization within the inferior parietal lobules. However, none of these additional analyses found evidence for an effect of alpha lateralization on neural speech tracking (see Supplementary Tables 8–11 and Supplementary Fig. 8).

Third, we asked whether our neural speech tracking results were impacted by the range of time lags used for reconstruction, or by the specific decoder model underlying the neural tracking index. Reconstructing envelopes using a shorter time window (50–250 ms) did not significantly change the resulting neural tracking index values (LMM, $\beta = 0.002$, SE = 0.007, $P = 0.84$; see also Supplementary Fig. 9). In a separate analysis, we calculated the neural tracking index using only the attended decoder model and probed its influence on behaviour. The results are overall in line with our main conclusions and particularly underscore the impact of neural speech tracking on response accuracy (see Supplementary Tables 14–17 for details).

Finally, we tested whether changes in age or hearing loss would modulate the relationship of neural tracking and alpha lateralization with listening behaviour. However, the inclusion of the respective interaction terms did not further improve the statistical models of accuracy and response speed (all $P$ values > 0.44).

## Discussion

We have utilized the power of a representative sample of middle-aged and older listeners to explicitly address the question of how two eminent neurobiological implementations of attentional filtering, typically studied in isolation, relate to one another, and how they jointly shape listening success. In addition, we leveraged our age-varying sample to ask how chronological age and hearing loss affect the fidelity of neural filter strategies and their influence on behaviour.

In our dichotic listening task, we source-localized the electro-encephalogram, and primarily focused on systematic spatial-cue-driven changes within the auditory cortex in alpha lateralization and the neural tracking of attended versus ignored speech. These results provide large-sample support for their suggested roles as neural instantiations of selective attention.

First, an informative spatial-attention cue not only boosted both neural measures but also consistently boosted behavioural performance. Listening behaviour was additionally influenced by both trial-to-trial and individual-to-individual variation in neural speech tracking, with relatively stronger tracking of the target sentence leading to better performance. An informative semantic cue led to faster responses but did not affect the two neural measures, thus most likely reflecting a priming effect speeding up the analysis of response alternatives rather than the differential processing of the sentences themselves.

Second, when related at the single-trial, single-subject level, the two neural attentional filter mechanisms were found to operate statistically independent of each other. This underlines their functional segregation and speaking to two distinct neurobiological implementations. Yet, when related in a coarser, between-subjects analysis across time, peaks in selective neural speech tracking systematically preceded those in alpha lateralization.

Importantly, while chronological age and hearing loss reliably decreased behavioural performance they did not systematically

affect the fidelity of neural filter strategies nor their influence on behaviour.

**Neural speech tracking but not alpha lateralization predicts listening success**. This study explicitly addressed the often overlooked question of how neural filter states (i.e., fluctuations from trial-to-trial) impact behaviour, here single-trial listening success[54–57]. Using a sophisticated linear-model approach that probed the impact of both state- and trait-level modulation of neural filters on behaviour, we only found evidence for a direct influence of neural speech tracking but not alpha lateralization on behavioural performance even though all three measures were robustly modulated by the presence of a spatial cue (see Fig. 6). What could be the reason for this differential impact of neural measures on behaviour?

To date, the behavioural relevance of selective neural speech tracking is still poorly supported given the emphasis on more naturalistic, yet more complex language stimuli[58,59]. While these stimuli provide a window onto the most natural forms of speech comprehension, they are not easily paired with fine-grained measures of listening behaviour. This makes it particularly challenging to establish a direct link between differential neural speech tracking and listening success[23,25,26,49,60]. Nevertheless, there is preliminary evidence linking stronger neural tracking to improved comprehension when it is tested at a comparably high level[28] (i.e., content questions on longer speech segments). Our current results thus provide important additional fine-grained and temporally resolved support to the functional relevance of selective neural speech tracking for moment-to-moment listening behaviour[61–63].

Despite a vast number of studies investigating the role of (lateralized) alpha oscillations in attentional tasks, the circumstances under which their top-down modulation may affect the behavioural outcome are still insufficiently understood[31]. Rather, the presence of a stable brain–behaviour relationship hinges on several factors.

First, the link of neural filter state to behaviour seems to be impacted by age: most evidence linking increased alpha lateralization to better task performance in spatial-attention tasks stems from smaller samples of young adults[12,15,64,65]. By contrast, the presence of such a systematic relationship in middle-aged and older adults is obscured by considerable variability in age-related changes at the neural (and to some extent also behavioural) level[14,66–69] (see discussion below).

Second, previous findings differ along (at least) two dimensions: (i) whether the functional role of alpha lateralization is studied during attention cueing, stimulus anticipation, or stimulus presentation[9,66,70], and (ii) whether the behaviour is related to the overall strength of alpha lateralization or its stimulus-driven rhythmic modulation[12,14]. Depending on these factors, the observed brain–behaviour relations may relate to different top-down and bottom-up processes of selective auditory attention.

Third, as shown in a recent study by Wöstmann et al.[70], the neural processing of target and distractor is supported by two uncorrelated lateralized alpha responses emerging from different neural networks. Notably, their results provide initial evidence for the differential behavioural relevance of neural responses related to target selection and distractor suppression, respectively.

In summary, it is still a matter of debate by which mechanistic pathway, and at which processing stage the modulation of alpha power will impact behaviour. While it is (often implicitly) assumed that alpha oscillations impact behaviour via modulation of neural excitability and thus early sensory processing, there is little evidence that shows a direct influence of alpha oscillation on changes in neural excitability and on subsequent behaviour[31,71].

Lastly, the increase in alpha lateralization around final-word presentation could at least partially reflect post-perceptual processes associated with response selection rather than the perceptual analysis itself[72]. The observed combined influence of neural tracking and alpha lateralization on response speed but not accuracy would seem compatible with such an interpretation (but see also ref. [73] for the combined influence of non-lateralized alpha power and neural speech tracking on intelligibility in a non-spatial listening task).

Taken together, our results underscore the impact of prioritized sensory encoding of relevant sounds via selective neural speech tracking on listening performance and highlight the difficulty in establishing a comparable link for a neural signature as multifaceted as alpha oscillations[74–76].

**Are fluctuations in lateralized alpha power and neural speech tracking functionally connected?** We investigated attention-related changes in two neural filter strategies that (i) involve neurophysiological signals operating at different frequency regimes, (ii) are assumed to support auditory attention by different neural mechanisms, and (iii) are typically studied in isolation[6,22]. Here, we found both neural filter strategies to be impacted by the same spatial-attention cue which afforded insights into their neurobiological dependence.

There is preliminary evidence, mostly from between-subjects analyses, suggesting that the two neural filter strategies may exhibit a systematic relationship[9,14,32,33,77]. How the two neural filter strategies may be connected mechanistically is thus still an open question. We here asked whether concurrent changes in neural filter states would imply a neural hierarchy in which alpha-driven controlled inhibition modulates the amplification of behaviourally relevant sensory information via selective neural speech tracking[78–80].

Our in-depth trial-by-trial analysis revealed independent modulation of alpha power and neural speech tracking. At the same time, in our exploratory between-subjects cross-correlation analysis we observed a systematic temporal delay with peaks in neural speech tracking leading those in alpha lateralization. While the direction and duration of this delay were closely in line with previous findings[12,14], at this coarser level of analysis, they speak against a hierarchy of neural processing in which lateralized alpha responses govern the differential neural tracking of attended versus ignored speech[81].

Our single-trial results are well in line with recent reports of independent variation in alpha-band activity and steady-state (SSR) or frequency-following responses (FFR) in studies of visual-spatial attention[30,31,82]. In addition, the inclusion of single-trial alpha lateralization as an additional training feature in a recent speech-tracking study failed to improve the decoding of attention[83]. The results from our most fine-grained single-trial level of analysis thus speak against a consistent, linear relationship of momentary neural filter states. Instead, we observed the co-existence of two complementary but seemingly independent neurobiological solutions to the implementation of auditory selective attention.

How can this finding be reconciled with findings from previous electrophysiological studies[9,32,33] pointing towards a functional trade-off between neurobiological attentional-filter mechanisms? And what could be an advantage to independent neural solutions for selective auditory attention?

Our between-subjects cross-correlation analysis appears to provide at least tentative support for a systematic relationship in which peaks in neural speech tracking precede those in alpha lateralization. A closer inspection of the group-level temporal modulation of neural measures throughout sentence presentation, however, reveals some important differences to previous results.

Whereas earlier studies reported an acyclic waxing and waning of neural entrainment and alpha power in response to rhythmic auditory stimulation[12,14,33], in this study, the two neural measures show different temporal dynamics: neural speech tracking gradually increases leading up to the final word, while alpha lateralization peaks at the sentence and final-word onset. The temporal dynamics of alpha lateralization, in particular, may point to the strategic intermittent engagement of spatial attention in line with task demands[48].

Do these differences in temporal dynamics of the two neural filters challenge the existence of a systematic single-trial brain–brain relationship? Yes, but they also point to a potential benefit of independent neural filter solutions. If the two neural measures of auditory attention were indeed functionally unconnected as suggested by the current results, they would allow for a wider range of neural filter state configurations to flexibly adapt to the current task demands and behavioural goals. The co-existence of two independent but complementary filter mechanisms operating either via the selective amplification of relevant or via the controlled inhibition of irrelevant sounds, enables different modes of auditory attention to serve a listener's goal in the face of complex real-life listening situations[19,20,84].

**Do age and hearing loss affect neural filter strategies?** The detrimental effects of increasing age and associated hearing loss on speech comprehension in noisy listening situations are well attested[85] and borne out by the current results. However, the extent to which the neural implementations of attentional filtering are affected by age and hearing loss, and in how far they may constitute neural markers of age-related speech comprehension problems, remains poorly understood[51].

As in a previous study on a subset of the current sample, we found the fidelity of alpha lateralization unchanged with age[14]. Other studies on auditory attention, however, have observed diminished and less sustained alpha lateralization for older compared to younger adults that were to some extend predictive of behaviour[66,86,87].

Our observation of preserved neural speech tracking across age and hearing levels only partially agree with earlier findings. They are in line with previous reports of differential neural tracking of attended and ignored speech for hearing-impaired older adults that mirrored the attentional modulation observed for younger or older normal-hearing adults[45–47,88]. As revealed by follow-up analysis (see Supplementary Tables 18 and 19), however, our data do not provide evidence for a differential impact of hearing loss on the neural tracking of attended or ignored speech as found in some of these studies. We also did not find evidence for overall increased levels of cortical neural tracking with age as observed in earlier studies[89,90].

The discrepancy in results may be explained by differences in (i) the studied populations (i.e., whether groups of younger and older participants were contrasted compared to the modelling of continuous changes in age and hearing loss), (ii) whether natural stories or short matrix sentence speech materials were used[91], or (iii) by differences in task details. In sum, the results suggest that the commonly observed adverse effects of age and hearing loss on speech-in-noise processing are not readily paired with concomitant changes at the neural level.

In a representative, age-varying sample of listeners, we underscore the functional significance of lateralized alpha power and neural speech tracking to spatial attention. Our results point to the co-existence of two independent yet complementary neural filter mechanisms to be flexibly engaged depending on a listener's attentional goals. However, we see no direct, behaviourally relevant impact of alpha-power modulation on early sensory gain processes.

Only for neural speech tracking, we established a mechanistic link from trial-to-trial neural filtering during the concurrent sound input to the ensuing behavioural outcome. This link exists irrespective of age and hearing status, which points to the potency of neural speech tracking to serve as an individualized marker of comprehension problems in clinical settings and as a basis for translational neurotechnological advances.

This key advance notwithstanding, the notable absence of an association between alpha lateralization and listening behaviour also highlights the level of complexity associated with establishing statistically robust relationships of complex neural signatures and behaviour in the deployment of auditory attention. To understand how the brain enables successful selective listening it is necessary that studies go beyond the characterization of neurobiological filter mechanisms alone, and further jointly account for the variability in both neural states and behavioural outcomes[92].

## Methods

**Data collection**. The analysed data are part of an ongoing large-scale study on the neural and cognitive mechanisms supporting adaptive listening behaviour in healthy middle-aged and older adults ('The listening challenge: How ageing brains adapt (AUDADAPT)'; https://cordis.europa.eu/project/rcn/197855_en.html). This project encompasses the collection of different demographic, behavioural, and neurophysiological measures across two time points. The analyses carried out on the data aim at relating adaptive listening behaviour to changes in different neural dynamics[35,36].

**Participants and procedure**. A total of $N = 155$ right-handed German native speakers (median age 61 years; age range 39–80 years; 62 males; see Supplementary Fig. 1 for age distribution) were included in the analysis. Handedness was assessed using a translated version of the Edinburgh Handedness Inventory[93]. All participants had normal or corrected-to-normal vision, did not report any neurological, psychiatric, or other disorders and were screened for mild cognitive impairment using the German version of the 6-Item Cognitive Impairment Test (6CIT[94]).

During the EEG measurement, participants performed six blocks of a demanding dichotic listening task (see Fig. 2 and Supplementary Methods for details on sentence materials).

As part of our the overarching longitudinal study on adaptive listening behaviour in healthy ageing adults, prior to the EEG session, participants also underwent a session consisting of a general screening procedure, detailed audiometric measurements, and a battery of cognitive tests and personality profiling (see ref. 14 for details). Only participants with normal hearing or age-adequate mild-to-moderate hearing loss were included (see Supplementary Fig. 1 for individual audiograms). As part of this screening procedure, an additional 17 participants were excluded prior to EEG recording due to non-age-related hearing loss or medical history. Three participants dropped out of the study prior to EEG recording and an additional nine participants were excluded from analyses after EEG recording: three due to incidental findings after structural MR acquisition, and six due to technical problems during EEG recording or overall poor EEG data quality. Participants gave written informed consent and received financial compensation (8€ per hour). Procedures were approved by the ethics committee of the University of Lübeck and were in accordance with the Declaration of Helsinki.

**Dichotic listening task**. In a recently established[35] linguistic variant of a classic Posner paradigm[34], participants listened to two competing, dichotically presented five-word sentences. They were probed on the sentence-final noun in one of the two sentences. All sentences followed the same sentence structure and had an average length of 2512 ms (range: 2183–2963 ms).

Sentences were spoken by the same female talker. Root mean (mean) square intensity (–26 dB Full Scale, FS) was equalized across all individual sentences and they were masked by continuous speech-shaped noise at a signal-to-noise ratio of 0 dB. Noise onset was presented with a 50 ms linear onset ramp and preceded sentence onset by 200 ms. Each sentence pair was temporally aligned by the onset of the two task-related sentence-final nouns. This, however, led to slight differences in the onset of the individual sentences. Crucially, the range and average sentence onset difference were similar for trials in which the probed (to-be-attended) sentence began earlier and those in which the unprobed (to-be-ignored) sentence began earlier (probed first: range: 0–580 ms, 162.1 ms ± 124.6; unprobed first: 0–560 ms, 180.6 ms ± 127.2). All participants listened to the same 240 sentence pairs but in subject-specific randomized order. In addition, across participants, we balanced the assignment of sentences to the right and left ear, respectively. Details on stimulus construction and recording can be found in the Supplementary Methods.

Critically, two visual cues preceded auditory presentation. First, a spatial-attention cue either indicated the to-be-probed ear, thus invoking selective attention, or did not provide any information about the to-be-probed ear, thus invoking divided attention. Second, a semantic cue specified a general or a specific semantic category for the final word of both sentences, thus allowing to utilize a semantic prediction. Cue levels were fully crossed in a 2 × 2 design and the presentation of cue combinations varied on a trial-by-trial level (Fig. 2a). The trial structure is exemplified in Fig. 2b.

Each trial started with the presentation of a fixation cross in the middle of the screen (jittered duration: mean 1.5 s, range 0.5–3.5 s). Next, a blank screen was shown for 500 ms followed by the presentation of the spatial cue in the form of a circle segmented equally into two lateral halves. In selective-attention trials, one half was black, indicating the to-be-attended side, while the other half was white, indicating the to-be-ignored side. In divided-attention trials, both halves appeared in grey. After a blank screen of 500 ms duration, the semantic cue was presented in the form of a single word that specified the semantic category of both sentence-final words. The semantic category could either be given at a general (natural vs. man-made) or specific level (e.g. instruments, fruits, furniture) and thus provided different degrees of semantic predictability. Each cue was presented for 1000 ms.

After a 500 ms blank-screen period, the two sentences were presented dichotically along with a fixation cross displayed in the middle of the screen. Finally, after a jittered retention period, a visual response array appeared on the left or right side of the screen, presenting four-word choices. The location of the response array indicated which ear (left or right) was probed. Participants were instructed to select the final word presented on the to-be-attended side using the touch screen. Among the four alternatives were the two actually presented nouns as well as two distractor nouns from the same cued semantic category. Note that because the semantic cue applied to all four alternative verbs, it could not be used to post hoc infer the to-be-attended sentence-final word.

Stimulus presentation was controlled by PsychoPy Standalone v2.0[95]. The visual scene was displayed using a 24" touch screen (ViewSonic TD2420) positioned within an arm's length. Auditory stimulation was delivered using in-ear headphones (EARTONE 3 A) at a sampling rate of 44.1 kHz. Following instructions, participants performed a few practice trials to familiarize themselves with the listening task. To account for differences in hearing acuity within our group of participants, individual hearing thresholds for a 500-ms fragment of the dichotic stimuli were measured using the method of limits. All stimuli were presented 50 dB above the individual sensation level. During the experiment, each participant completed 60 trials per cue–cue condition, resulting in 240 trials in total. The cue conditions were equally distributed across six blocks of 40 trials each (~10 min) and were presented in random order. Participants took short breaks between blocks.

**Behavioural data analysis.** We evaluated participants' behavioural performance in the listening task with respect to accuracy and response speed. For the binary measure of accuracy, we excluded trials in which participants failed to answer within the given 4-s response window ('timeouts'). Spatial stream confusions, that is trials in which the sentence-final word of the to-be-ignored speech stream were selected, and random errors were jointly classified as incorrect answers. The analysis of response speed, defined as the inverse of reaction time, was based on correct trials only. Single-trial behavioural measures were subjected to (generalized) linear mixed-effects analysis and regularized regression (see 'Statistical analysis').

**EEG data analysis.** The preprocessed continuous EEG data (see Supplementary Methods for details on data collection and preprocessing) were high-pass-filtered at 0.3 Hz (finite impulse response (FIR) filter, zero-phase lag, order 5574, Hann window) and low-pass filtered at 180 Hz (FIR filter, zero-phase lag, order 100, Hamming window). The EEG was cut into epochs of −2 to 8 s relative to the onset of the spatial-attention cue to capture cue presentation as well as the entire auditory stimulation interval.

For the analysis of changes in alpha power, EEG data were downsampled to $f_s$ = 250 Hz. Spectro-temporal estimates of single-trial data were then obtained for a time window of −0.5 to 6.5 s (relative to the onset of the spatial-attention cue) at frequencies ranging from 8 to 12 Hz (Morlet's wavelets; the number of cycles = 6).

For the analysis of the neural encoding of speech by low-frequency activity, the continuous preprocessed EEG were downsampled to $f_s$ = 125 Hz and filtered between $f_c$ = 1 and 8 Hz (FIR filters, zero-phase lag, order: $8 f_s/f_c$ and $2 f_s/f_c$, Hamming window). The EEG was cut to yield individual epochs covering the presentation of auditory stimuli, beginning at noise onset until the end of the auditory presentation.

Following EEG source and forward model construction (see Supplementary Methods for details), sensor-level single-trial data in each of our two analysis routines were projected to source space by matrix multiplication of the spatial filter weights. To increase the signal-to-noise ratio in source estimates and computationally facilitate source-level analyses, source-projected data were averaged across grid points per cortical area defined according to the HCP functional parcellation template[96,97]. This parcellation provides a symmetrical delineation of each hemisphere into 180 parcels for a total of 360 parcels. We constrained the analysis of neural measures to an a priori defined, source-localized auditory region of interest (ROI) as well as one control ROI in the inferior parietal

lobule (see Supplementary Methods for details). The described analyses were carried out using the Fieldtrip toolbox (v. 2017-04-28) in Matlab 2016b, and the Human Connectome Project Workbench software (v1.5) as well as FreeSurfer (v.6.0).

**Attentional modulation of alpha power.** Absolute source power was calculated as the square amplitude of the spectro-temporal estimates. Since oscillatory power values typically follow a highly skewed, non-normal distribution, we applied a nonlinear transformation of the Box-Cox family ($power_{trans} = (power^P − 1)/P$ with $P = 0.5$) to minimize skewness and to satisfy the assumption of normality for parametric statistical tests involving oscillatory power values[98]. To quantify attention-related changes in 8–12 Hz alpha power, per ROI, we calculated the single-trial, temporally resolved alpha lateralization index as follows[12,14,15]:

$$\text{ALI} = (\alpha\text{-power}_{ipsi} - \alpha\text{-power}_{contra})/(\alpha\text{-power}_{ipsi} + \alpha\text{-power}_{contra}) \quad (1)$$

To account for overall hemispheric power differences that were independent of attention modulation, we first normalized single-trial power by calculating per parcel and frequency the whole-trial (−0.5–6.5 s) power averaged across all trials and subtracted it from single trials. We then used a robust variant of the index that applies the inverse logit transform [$(1/(1 + \exp(−x))$] to both inputs to scale them into a common, positive-only [0;1]-bound space prior to index calculation.

For visualization and statistical analysis of cue-driven neural modulation, we then averaged the ALI across all parcels within the auditory ROI and extracted single-trial mean values for the time window of sentence presentation (3.5–6.5 s), and treated them as the dependent measure in linear mixed-effects analysis (see 'Statistical analysis' below). They also served as continuous predictors in the statistical analysis of brain–behaviour and brain–brain relationships. We performed additional analyses that focused on the ALI in the auditory cortex during presentation of the sentence-final word and spatial-attention cue, respectively. Further control analyses included single-trial ALI during sentence and final-word presentation that were extracted from the inferior parietal ROI.

**Estimation of envelope reconstruction models.** To investigate how low-frequency (i.e., <8 Hz) fluctuations in EEG activity related to the encoding of attended and ignored speech, we trained stimulus reconstruction models (also termed decoding or backward models) to predict the onset envelope (see Supplementary Methods for details) of the attended and ignored speech stream from EEG[99,100]. In this analysis framework, a linear reconstruction model $g$ is assumed to represent the linear mapping from the recorded EEG signal, $r(t,n)$, to the stimulus features, $s(t)$:

$$\hat{s}(t) = \sum_n \sum_\tau g(\tau, n) r(t + \tau, n) \quad (2)$$

where $\hat{s}(t)$ is the reconstructed onset envelope at time point $t$. We used all parcels within the bilateral auditory ROI and time lags $\tau$ in the range of −100 ms to 500 ms to compute envelope reconstruction models using ridge regression[101]:

$$g = (R^T R + \lambda m I)^{-1} R^T s \quad (3)$$

where R is a matrix containing the sample-wise time-lagged replication of the neural response matrix r, $\lambda$ is the ridge parameter for regularization, I is the identity matrix, and $m$ is a subject-specific scalar representing the mean of the trace of $R^T R$[102,103]. The same grid of ridge parameters ($\lambda = 10^{-5}, 10^{-4}, …10^{10}$) was used across subjects, and $m$ proved to be relatively stable across subjects ($387.2 \pm 0.18$, mean ± SD). The optimal ridge value of $\lambda = 1$ was determined based on the average Pearson's correlation coefficient and mean squared error of the reconstructed and actually presented envelope across all trials and subjects.

Compared to linear forward ('encoding') models that derive temporal response functions (TRFs) independently for each EEG channel or source, stimulus reconstruction models represent multivariate impulse response functions that exploit information from all time lags and EEG channels/sources simultaneously. To allow for a neurophysiological interpretation of backward model coefficients, we additionally transformed them into linear forward model coefficients[104]. All analyses were performed using the multivariate temporal response function (mTRF) toolbox[99] (v1.5) for Matlab (v2016b).

Prior to model estimation, we split the data based on the two spatial-attention conditions (selective vs. divided), resulting in 120 trials per condition. Envelope reconstruction models were trained on concatenated data from selective-attention trials, only. Prior to concatenation, single trials were zero-padded for 600 ms to reduce discontinuity artefacts, and one trial was left out for subsequent testing. On each iteration, two different backward models were estimated, an envelope reconstruction model for the-be-attended speech stream (short: attended reconstruction model), and one for the to-be-ignored speech stream (short: ignored reconstruction model). Reconstruction models for attended and ignored speech signals were trained separately for attend-left and attend-right trials which yielded 120 decoders (60 attended, 60 ignored) per attentional setting. For illustrative purposes, we averaged the forward-transformed models of attended and ignored speech per hemisphere across all participants (Fig. 4b).

**Evaluation of neural tracking strength**. We analysed how strongly the attended compared to ignored sentences were tracked by slow cortical dynamics by quantifying the envelope reconstruction accuracy for individual trials. To this end, we reconstructed the attended and ignored envelope of a given trial using a leave-one-out cross-validation procedure. The two envelopes of a given trial were reconstructed using the models trained on all but the current trial from the same attention condition. The reconstructed onset envelope obtained from each model was then compared to onset envelopes of the actually presented speech signals using a 248-ms sliding window (rectangular window, step size of 1 (8 ms) sample). The resulting time courses of Pearson correlation coefficients, $r_{attended}$ and $r_{ignored}$, reflect a temporally resolved measure of single-trial neural tracking strength or reconstruction accuracy[28] (see Fig. 4 and Supplementary Fig. 4).

We proceeded in a similar fashion for divided-attention trials. Since these trials could not be categorized based on the to-be-attended and -ignored sides, we split them based on the ear that was probed at the end of the trial. Given that even in the absence of a valid attention cue, participants might still (randomly) focus their attention on one of the two streams, we wanted to quantify how strongly the probed and unprobed envelopes were tracked neurally. We used the reconstruction models trained on selective-attention trials to reconstruct the onset envelopes of divided-attention trials. Sentences presented in probed-left/unprobed-right trials were reconstructed using the attend-left/ignore-right reconstruction models while probed-right/unprobed-left trials used the reconstruction models trained on attend-right/ignore-left trials.

**Attentional modulation of neural tracking**. In close correspondence to the alpha lateralization index, we calculated a neural tracking index throughout sentence presentation. The index expresses the difference in neural tracking of the to-be-attended and ignored sentence (in divided attention: probed and unprobed, respectively)[27]:

$$\text{Neural tracking index} = (r_{attended} - r_{ignored})/(r_{attended} + r_{ignored}) \quad (4)$$

Positive values of the resulting index indicate that the attended envelope was tracked more strongly than the ignored envelope, and vice versa for negative values. Since individual sentences differed in length, for visualization and statistical analysis, we mapped their resulting neural tracking time courses onto a common time axis expressed in relative (percent) increments between the start and end of a given stimulus. We first assigned each sample to one of 100 bins covering the length of the original sentence in 1% increments. We then averaged across neighbouring bins using a centred rectangular 3% sliding window (1% overlap). The same procedure was applied to the time course of alpha-power lateralization following up-sampling to 125 Hz. Single-trial measures for the interval of final-word presentation were averaged across the final 35% of sentence presentation as this interval covered final-word onset across all 240 sentence pairs. We used the single-trial neural tracking index as (in-)dependent variables in our linear mixed-effects analyses (see below).

**Statistical analysis**. We used (generalized) linear mixed-effect models to answer the research questions outlined in Fig. 1. This approach allowed us to jointly model the impact of listening cues, neural filter strategies and various additional covariates known to influence behaviour. These included the probed ear (left/right), whether the later-probed sentence had the earlier onset (yes/no), as well as participants' age and hearing acuity (pure-tone average across both ears).

To arbitrate between state-level (i.e., within-subject) and trait-level (i.e., between-subject) effects, our models included two separate regressors for each of the key neural measures. Between-subject effect regressors consisted of neural measures that were averaged across all trials at the single-subject level, whereas the within-subject effect was modelled by the trial-by-trial deviation from the subject-level mean[50].

Deviation coding was used for categorical predictors. All continuous variables were z-scored. For the dependent measure of accuracy, we used a generalized linear mixed-effects model (binomial distribution, logit link function). For response speed, we used a general linear mixed-effects model (Gaussian distribution, identity link function). Given the sample size of $N = 155$ participants, $P$ values for individual model terms are based on Wald $t$-as-$z$-values for linear models[105] and on z-values and asymptotic Wald tests in generalized linear models. All reported $P$ values are corrected to control for the false discovery rate at $q = 5\%$[106].

In lieu of a standardized measure of effect size for mixed-effects models, we report odds ratios (OR) for generalized linear models and standardized regression coefficients (β) for linear models along with their respective standard errors (SE).

All analyses were performed in R (v3.6.1)[107] using the packages lme4 (v1.1-23)[108], and sjPlot (v2.8.5)[109].

**Model selection**. To avoid known problems associated with a largely data-driven stepwise model selection that includes the overestimation of coefficients[110] or the selection of irrelevant predictors[111], the inclusion of fixed effects was largely constrained by our a priori defined hypotheses. The influence of visual cues and of neural measures was tested in the same brain–behaviour model. The brain–behaviour model of accuracy and response speed included random intercepts by subject and item. In a data-driven manner, we then tested whether model fit could be further improved by the inclusion of subject-specific random slopes for the effects of the spatial-attention cue, semantic cue, or probed ear. The change in model fit was assessed using likelihood ratio tests on nested models.

**Reporting summary**. Further information on research design is available in the Nature Research Reporting Summary linked to this article.

## Data availability
The complete neural and behavioural data required to reproduce the analyses supporting this work, as well as the auditory stimuli used in this study are publicly available in the study's Open Science Framework repository (https://osf.io/nfv9e/). Source data are provided with this paper.

## Code availability
Code for the analyses supporting this work is publicly available in the study's Open Science Framework repository (https://osf.io/nfv9e/).

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

## Acknowledgements

The research was funded by the European Research Council (grant no. ERC-CoG-2014-646696"Audadapt" awarded to J.O.). The authors are grateful for the help of Franziska Scharata in acquiring the data.

## Author contributions

S.T., M.A. and J.O. designed the experiment; S.T. and M.A. oversaw data collection and preprocessing of the data; S.T., M.A. and L.F. analysed the data under supervision by J.O.; S.T. wrote the first draft; all authors contributed to the final version of the manuscript.

## Funding

## Competing interests

The authors declare no competing interests.
