## [Peer Review File · Nature Communications]

REVIEWER COMMENTS

Reviewer #1 (Remarks to the Author):

This manuscript presents research aimed at examining how two established electrophysiological signatures of selective attention to speech relate to behavior and to each other in a large sample of adults of varying age. The authors recorded EEG from 155 subjects (aged 39-80) while they (dichotically) presented them with two (more or less) concurrent speech streams. In advance, they cued the subjects to attend to one stream based on a spatial cue or they cued them with some predictive information about the semantic category of the final (target) word in the two streams. They then analyzed the EEG data to derive two known signatures of attention (neural tracking of the speech envelope & alpha power lateralization). They report a number of findings: they find robust neural tracking and alpha lateralization across age, they find neural speech tracking and alpha lateralization scores do not correlate with each other across trials or subjects, and they report that neural tracking better correlates with trial-to-trial behavior.

Overall, this is a well-written manuscript on an interesting topic that asks a variety of questions using high quality data and analyses, and with a reasonable discussion. Nonetheless, I have a few comments and queries for the authors.

Main comments:

- 1) I found the whole idea of the semantic cue a little bit strange. It wasn't clear why you used it or what you expected to find. It might be worth adding a bit of motivation to the introduction on that – even if the results didn't work out the way you expected. At the moment, it is just introduced and never really discussed.
- 2) I wondered about the decision to explore the alpha lateralization (and neural tracking even) across the entire timeframe of the sentences. Given that subjects just need to report the final (target) word – and that the target words don't seem to be semantically predictable from the previous words in the stimulus – I wondered would subjects strategically deploy their attention only toward the end of the sentences. Indeed, this might be what is going on in figure 3, and figure 4. They may be orienting to the target after the cue... then relaxing... then orienting again?
- 3) Related to the previous point, I wonder might it be worth linking to some work by Barb Shinn-Cunningham's group in which they suggest that alpha lateralization might be involved in orienting attention spatially, but once attention is deployed to the auditory object, it is no longer necessary to use space to maintain it (Bonacci et al., 2020). That might partially explain the dissociation between alpha and tracking here?
- 4) The final paragraph of the introduction stood out to me as representing one other limitation of the present work. Specifically, it focuses so narrowly on alpha and neural tracking. These are worthy of study, no question. And I am not suggesting you need to broaden the analysis at all. But the fact remains that there are other measures of neural "tracking" that have recently been shown to be much more sensitive to attention – including some that correlate quite strongly with behavior (e.g., Brodbeck et al., 2018, Broderick et al., 2018). Again, not suggesting you need to include these kinds of measures. Just that it might be worth mentioning that alpha and envelope tracking are not the only way to index "neural filters".
- 5) On that note – I believe O'Sullivan et al., 2014 also showed a correlation between neural tracking and behavior across trials.
- 6) Another thought I had when looking at the results in figure 4 was about the decision to use attended decoders to decode attended speech and unattended to decode

unattended speech. This is totally legitimate in my view. But it doesn't have to be the only and it may not even be the best way to distinguish attended from unattended. (This is because – in the limit of perfect SNR – one might expect attended and unattended decoding to both be perfect; an unattended decoder is not just a weaker attended decoder – Ding & Simon, 2012). I know some papers have used an attended decoder to decode both attended and unattended speech – with a view to maximizing the sensitivity to the attention effect. I think is legitimate too (depending on the particulars of the experiment – it can be biased in some cases). But it would seem reasonable here. So – if you think it might be worth it – you might consider an analysis like that to see if a neural tracking index derived in that way might correlate even more strongly with behavior. Up to you though.

Minor comments:

- 1) You use the phrase “alternating regimes” in the significance statement – this could be read as implying that the tracking and alpha lateralization wax and wane in an alternating manner with “versus” each other – which is not true, right? They are independent. Maybe, rephrase for precision?
- 2) “synoptic”? Is that what you mean to say? As in a “summary” look?? Doesn't seem to be the right word to me. Maybe “synergistic”?
- 3) I was a bit unclear on lines 448-450 – did you mean specifically elevated alpha with reduced tracking? Or vice versa? Or it can go both ways?
- 4) Using time lags from -100 to 500 ms for the neural tracking is definitely sub optimal. You might have gotten improved sensitivity if you had used something like 50 – 250 ms (i.e., where there is actual signal).

Typos:

Subtitle on line 411 – “power” not “lower”

Reviewer #2 (Remarks to the Author):

This study addresses neural measures of attentive listening to speech, in a large group of adults who span a wide range of age (39-80 years) and typical hearing loss. In recent years, two neural correlates of attending to speech in noise, commonly measured with EEG or MEG, have become increasingly recognized: modulation of alpha power lateralization between hemispheres and modulation of low-frequency speech envelope tracking. However, despite ample evidence of their involvement during attentive listening, there is little evidence relating these neural measures to each other or to their joint functional consequences behaviorally. The present study tackles this important, outstanding mechanistic question in the field. The authors take a remarkably comprehensive approach, incorporating how the nature of the attentional cue (spatial vs semantic) affects the neural metrics, and how the neural metrics relate to one another, to which ear is doing the listening, to behavior, and to other demographic variables (age, hearing loss). Furthermore, the study does all this in a well balanced manner, both across subjects and across individual trials, and over time within trials. It's beautifully, thoughtfully designed. Given all the experimental factors, the results are complex, but they basically demonstrate that spatially (but not semantically) informative attentional cueing increases both alpha lateralization and speech tracking in presumed auditory cortices. Importantly, alpha lateralization is largely uncorrelated with speech tracking strength (particularly on an individual trial basis, and regardless of spatial attention), and only speech tracking helps explain improved comprehension. Alpha lateralization and speech tracking thus appear to reflect somewhat independent attentional mechanisms, with alpha showing no behavioral impact in this task – a somewhat surprising result that holds

across age and hearing status.

As noted above, this study addresses a deep, fundamental question in the field, and it is likely broadly relevant to other modalities too (e.g. vision). Methodologically it is well crafted, well executed, and utterly meticulous: the cohort is very large and well characterized, stimuli and tasks are cleverly designed, analyses directly address the hypotheses, good control analyses are run, and statistics are thorough and appropriate. The data and analysis code will evidently be posted upon publication, which will be a tremendous boon to the field and will support replication efforts. The main challenge is that this study raises more questions than it answers, and the reader is still left wondering about the original motivation: as the authors put it in the discussion, “how do the two neural filters relate to one another, and how do they influence listening success in a demanding real-life listening situation?” On this point, the reader would benefit from a clearer take-home message, even if it’s somewhat speculative.

MAJOR

It is unclear how to interpret the rich results. Examples include: If alpha lateralization and speech tracking are “independent” mechanisms, what distinct roles could they be serving? Other investigators have observed a relation between alpha lateralization and performance; why don’t we see it here? Why is semantically informed attention so ineffective in eliciting a neural effect, even while improving behavior – doesn’t that call into question the directness of the causal relationship between tracking and comprehension? As I mention above, the reader could use much more help contextualizing these results within the original motivation.

MINOR

- 1) I was confused when alpha and speech tracking are termed neural or auditory “filtering strategies”. Elsewhere you use clearer terms such as “neurobiological measures of auditory attention” or “instantiation of attention at the neurobiological level”.
- 2) You claim that since alpha lateralization and tracking vary independently, this demonstrates a “functional trade-off”. Even with the grand average temporal relationship (tracking preceding alpha), that’s rather hard to prove given the stereotyped trial structure.
- 3) The alpha that shows no relationship to tracking happens to peak at the end of the final word, which suggests that it could have as much to do with the subsequent remembering and responding rather than comprehending. This fits with the fact that final-word alpha did relate (jointly with neural tracking) to response speed but not accuracy. Kindly comment on this possibility.

Reviewer #3 (Remarks to the Author):

The manuscript by Tune et al represents a behemoth of a study examining both neural tracking and alpha oscillations in aging subjects in a challenging listening task. The manuscript suggests that attention-driven neural tracking and alpha modulation are independent processes not affected by age.

The manuscript is well written, easily to understand and the results are transparent as evidenced by the supplement data. I find no issues with the statistical methods used by the authors – they are very appropriate for this line of research.

A major finding was that neural tracking and alpha were independent is surprising. A “nice/simple” story would be that alpha primes the auditory cortex to be more sensitive to neural tracking .. this doesn’t appear to be the case. Although the authors mention Hauswald et al.’s paper that a joint model with alpha and speech tracking better predicted behaviour than

either alone, some elaboration on this discrepancy would be beneficial.

I find very little room for any criticism. Perhaps only minor clarification:

-The use of single trial neural tracking has not been well explored in the past and the use of this method for alpha single trial correlations is insightful. How were the "sliding windows" implemented? i.e., shape, overlap...

What were the range of lamdas (in the TRF script) that were finally used in the neural tracking measure? Was the same lambda used for everyone or was it optimized per subject?

-The use of auditory cortex ROI for the alpha was a bit surprising given that a lot of previous work found dominant generator in the parietal regions for auditory spatial processing.

However, the use of a parietal control ROI puts my mind at ease. Were the responses at parietal larger/more robust than auditory cortex?

Andrew Dimitrijevic

Reviewer #1 (Remarks to the Author):

This manuscript presents research aimed at examining how two established electrophysiological signatures of selective attention to speech relate to behavior and to each other in a large sample of adults of varying age. The authors recorded EEG from 155 subjects (aged 39-80) while they (dichotically) presented them with two (more or less) concurrent speech streams. In advance, they cued the subjects to attend to one stream based on a spatial cue or they cued them with some predictive information about the semantic category of the final (target) word in the two streams. They then analyzed the EEG data to derive two known signatures of attention (neural tracking of the speech envelope & alpha power lateralization). They report a number of findings: they find robust neural tracking and alpha lateralization across age, they find neural speech tracking and alpha lateralization scores do not correlate with each other across trials or subjects, and they report that neural tracking better correlates with trial-to-trial behavior.

Overall, this is a well-written manuscript on an interesting topic that asks a variety of questions using high quality data and analyses, and with a reasonable discussion. Nonetheless, I have a few comments and queries for the authors.

We thank the reviewer for their positive and constructive review. Perhaps most importantly, based on the points raised, we carried out two control analyses for neural speech tracking that lend additional credibility to the originally reported results and conclusions. Please see below for a detailed point-by-point response to all concerns.

Main comments:

1) I found the whole idea of the semantic cue a little bit strange. It wasn't clear why you used it or what you expected to find. It might be worth adding a bit of motivation to the introduction on that – even if the results didn't work out the way you expected. At the moment, it is just introduced and never really discussed.

We thank the Reviewer for this helpful comment. As Reviewer # 2 raised a related concern on the effect (or lack thereof) of the semantic cue on behaviour and on the two neural measures of interest, we jointly respond to both of these points here.

Overall, we acknowledge that the way we introduced and discussed the effects of the semantic cue may have caused more confusion for the reader than it has added insight.

To adjust the reader's expectations, the revised introduction now states more clearly that the main focus of the presented analysis laid on the neurobiological implementations of selective attention, and that the semantic cue did indeed play a secondary role for the set of research questions addressed. As the current study is part of an ongoing large-scale project on the neural and cognitive mechanisms supporting adaptive listening behaviour in healthy middle-aged and older adults, we primarily included the semantic cue to investigate whether ageing would change the relative importance of cognitive strategies encouraged by the two different cues. However, our predictions were not borne out by the data and since the semantic cue effect is only marginally connected to the main focus of the current manuscript, the lack of its effect on neural measures is reported for the sake of transparency but its role for the overall narrative of the manuscript is toned down accordingly.

The relevant introduction section now reads as follows:

"In the present EEG study that is part of an ongoing large-scale project on the neural and cognitive mechanisms supporting adaptive listening behaviour in healthy aging, we aim at closing these gaps by leveraging the statistical power and representativeness of our large, age-varying participant sample. We use a novel dichotic listening paradigm to enable a synergistic look at concurrent changes in

auditory alpha power and neural speech tracking at the single-trial level. More specifically, our linguistic variant of a classic Posner paradigm (Posner, 1980) emulates a challenging dual-talker listening situation in which speech comprehension is supported by two different listening cues (Alavash, Tune, & Obleser, 2019). These cues encourage the use of two complementary cognitive strategies to improve comprehension: Of particular interest for the present scientific endeavour is the spatial-attention cue that guides auditory attention in space. We additionally manipulated semantic predictability of upcoming speech via a semantic category cue. While the effects of the semantic cue are of secondary importance for the present research questions, its manipulation still allows insights into how it affects listening success in a large cohort of middle-aged and older adults. As previous research has shown that the sensory analysis of speech and, to a lesser degree, the modulation of alpha power are influenced by the availability of higher-order linguistic information, we also investigated whether semantic predictability would modulate neural tracking and alpha lateralization (Broderick, Anderson, & Lalor, 2019; Obleser & Weisz, 2012; Peelle, Gross, & Davis, 2013; Presacco, Simon, & Anderson, 2016; Sohoglu, Peelle, Carlyon, & Davis, 2012; Wöstmann, Lim, & Obleser, 2017)."

With respect to semantic cue effect on response speed but not on neural measures, we would argue that the relatively faster responses following a specific semantic cue are most likely caused by a semantic priming effect that led to shorter reading times for the four answer options. In other words, a specific semantic cue probably had a stronger impact on the analysis of response alternative in the decision process than on the processing of the sentences themselves which could also account for the absence of a semantic effect on the two neural measures. While we had already offered an explanation along those lines in the results section of the original manuscript, in the revised version, we additionally contextualize these results in the first paragraph of the discussion as well:

"In our dichotic listening task, we source-localized the electroencephalogram, and primarily focused on the systematic spatial-cue-driven changes within auditory cortex in alpha power lateralization and the differential neural tracking of attended versus ignored speech.

[...]

An informative semantic cue led to faster responses but did not affect the two neural measures, thus most likely reflecting a priming effect speeding up the analysis of response alternatives rather than a differential processing of the sentences themselves."

In the results section, we now also make more explicit that our expectation of more accurate responses following an informative semantic cue was not borne out by the data:

"Also, participants responded generally faster in trials in which they were given a specific, more informative semantic cue (LMM; $\beta=.2$, $SE=.03$, $p<.001$), most likely reflecting a semantic priming effect that led to faster word recognition. Contrary to our expectations, a more informative semantic cue did not lead to more accurate responses (GLMM; $OR=1.1$, $SE=.11$, $p=.69$)."

2) I wondered about the decision to explore the alpha lateralization (and neural tracking even) across the entire timeframe of the sentences. Given that subjects just need to report the final (target) word – and that the target words don't seem to be semantically predictable from the previous words in the stimulus – I wondered would subjects strategically deploy their attention only toward the end of the sentences. Indeed, this might be what is going on in figure 3, and figure 4. They may be orienting to the target after the cue... then relaxing... then orienting again?

We agree with the reviewer that the period around final word presentation is particularly relevant for behavioural performance and thus of interest for the analysis of neural measures. For this reason, all analyses involving neural speech tracking or alpha lateralization were carried out on both *the entire sentence* and *for the final word period only*.

Our rationale for also modelling changes in neural speech tracking and alpha lateralization *across the entire sentence presentation* was motivated by the characteristics of our experimental design, as well as by previous results that showed temporally modulated alpha lateralization throughout auditory presentation of similar duration (e.g., Wöstmann et al. 2016, PNAS; Tune et al. 2018, Eur J Neurosci).

With respect to the design, it is important to realize that both sentences were spoken by the same female talker and followed a stereotyped, semantically non-predictive structure. In the absence of additional cues such as speaker identity or contextual information to guide attention, we reasoned that listeners would use the entire sentence context leading up to the final word to ‘tune-in’ to and keep track of the to-be-attended sentence. In addition, semantic cue presentation prior to sentence presentation could perturb attentional orientation which would require a re-orientation of attention at sentence onset that could be predictive of listening success.

These assumptions were in part borne out by the data: on the one hand, we observed an increase in alpha lateralization not only during the final word but also at the beginning of sentence presentation. On the other hand, we found that neural tracking was increased for sentences that started ahead of the competing sentence, thus pointing to the relevance of the entire sentence presentation period for the engagement of attention.

Throughout the results section of the revised manuscript, we now more clearly state the expected pattern and point the reader to the separate but analogous analyses of the two time-windows of interest (see Supplementary Tables 6–8 and 12–13).

3) Related to the previous point, I wonder might it be worth linking to some work by Barb Shinn-Cunningham’s group in which they suggest that alpha lateralization might be involved in orienting attention spatially, but once attention is deployed to the auditory object, it is no longer necessary to use space to maintain it (Bonacci et al., 2020). That might partially explain the dissociation between alpha and tracking here?

We thank the reviewer for bringing this relevant publication to our attention. In line with our response to the previous comment, a main difference to the experimental design by Bonacci et al. 2020 is that our task did not offer any additional features (such as pitch) that could help to maintain the focus of attention throughout sentence presentation.

Nevertheless, as the reviewer rightly stated above, our alpha results show intermittent rather than sustained periods of strong lateralization, suggesting that listeners strategically upregulated their attention during critical parts of the trial. While we also observed a similar increase in neural tracking around the final word, this increase was more gradual across sentence presentation. These differences in the temporal dynamics of neural measures may have indeed contributed to their statistical independence observed in the current study.

We now do cite the Bonacci et al. paper in the according section of the discussion (see below). A relevant quote from the revised discussion reads as follows:

“Our between-subjects cross-correlation analysis appears to provide at least tentative support for a systematic relationship in which peaks in neural tracking precede those in alpha power lateralization. A closer inspection of the group-level temporal modulation of neural measures throughout sentence presentation, however, reveals some important differences to previous results. Whereas those studies reported an acyclic waxing and waning of neural entrainment and alpha power at frequency of a much more rhythmic stimulation (Lakatos et al., 2016; Tune, Wöstmann, & Obleser, 2018; Wöstmann, Herrmann, Maess, & Obleser, 2016), in the current study, the two neural measures exhibit different temporal dynamics: neural tracking gradually increases leading up to the final word, while alpha lateralization peaks at sentence and final word onset. The temporal dynamics of alpha lateralization,

in particular, may point to strategic intermittent engagement of spatial attention in line with task demands (Bonacci, Bressler, & Shinn-Cunningham, 2020). “

4) The final paragraph of the introduction stood out to me as representing one other limitation of the present work. Specifically, it focuses so narrowly on alpha and neural tracking. These are worthy of study, no question. And I am not suggesting you need to broaden the analysis at all. But the fact remains that there are other measures of neural “tracking” that have recently been shown to be much more sensitive to attention – including some that correlate quite strongly with behavior (e.g., Brodbeck et al., 2018, Broderick et al., 2018). Again, not suggesting you need to include these kinds of measures. Just that it might be worth mentioning that alpha and envelope tracking are not the only way to index “neural filters”.

We have addressed this important comment together with Minor comment #1 by Reviewer 2 stated below. In responding to the related concerns regarding our introduction of ‘neural filters’, we now more clearly define and motivate the term as an important algorithmic metaphor and emphasize that alpha lateralization and neural tracking of the speech envelope are only two potential but nonexclusive neurobiological solutions.

The relevant section of the introduction now reads as follows:

[...]

“Successful speech comprehension thus relies on the differential treatment of relevant and irrelevant inputs. Here, the idea of neural attentional ‘filters’ serves as an important and pervasive algorithmic metaphor of how attention is implemented at the neural level (Broadbent, 1958; Fernandez-Duque & Johnson, 1999; Obleser & Erb, 2020). Recent neuroscientific studies on the neurobiological mechanisms supporting the controlled inhibition and amplification of speech have predominantly focused on two potential but nonexclusive neural filter strategies originating from distinct research traditions:

[...]

Slow cortical dynamics temporally align with (or “track”) auditory input signals to prioritize the neural representation of behaviourally-relevant sensory information (Henry & Obleser, 2012; Obleser & Kayser, 2019; Schroeder & Lakatos, 2009; Schroeder, Wilson, Radman, Scharfman, & Lakatos, 2010); but see also Brodbeck et al., 2018; Broderick et al., 2018 for the neural tracking of contextual semantic information).”

5) On that note – I believe O’Sullivan et al., 2014 also showed a correlation between neural tracking and behavior across trials.

We apologize for this oversight. We have added a reference to this study to the introduction section where we discuss the sparsity of evidence that links neural measures of selective attention to behaviour.

6) Another thought I had when looking at the results in figure 4 was about the decision to use attended decoders to decode attended speech and unattended to decode unattended speech. This is totally legitimate in my view. But it doesn’t have to be the only and it may not even be the best way to distinguish attended from unattended. (This is because – in the limit of perfect SNR – one might expect attended and unattended decoding to both be perfect; an unattended decoder is not just a weaker attended decoder – Ding & Simon, 2012). I know some papers have used an attended decoder to decode both attended and unattended speech – with a view to maximizing the sensitivity to the attention effect. I think is legitimate too (depending on the particulars of the experiment – it can be biased in some cases). But it would seem reasonable here. So – if you think it might be worth it – you might consider an analysis like that to see if a neural tracking index derived in that way might correlate even more strongly with behavior. Up to you though.

We thank the reviewer for pointing us to this alternative analysis approach. In the original version of the analysis, we had intentionally based the neural tracking index on both the attended and ignored decoder to incorporate information about how the brain treats both relevant and irrelevant auditory input. This also ensured that the analyses of both alpha lateralization and neural speech tracking were kept as analogous as possible.

We agree, however, that the approach proposed by the reviewer could potentially help to maximize the effect of neural tracking on behaviour, if one keeps in mind that it is primarily based on neural responses to the to-be-attended input.

Given the use of this alternative approach in previous studies, we ran a corresponding control analysis (see Supplementary Tables 14–17) that lend additional credibility to our conclusions. In short, the results confirmed that stronger neural tracking of attended vs. ignored inputs (here particularly during the final word and to a lesser degree during the entire sentence presentation) led to overall more accurate responses.

The relevant section on control analyses now reads as follow. Note that we also report the results of an additional control analysis that addressed the reviewer’s minor comment on the chosen time lags:

“We ran additional control analyses to validate our main set of results.

[...]

Third, we asked whether our neural tracking results were impacted by the range of time lags used for reconstruction, or by the specific decoder model underlying the neural tracking index. Reconstructing envelopes using a shorter time window (50–250 ms) did not significantly change the resulting neural tracking index values (LMM, $\beta = .002$, $SE = .007$, $p = .84$; see also Supplementary Fig. 9).

In a separate analysis, we calculated the neural tracking index using only the attended decoder model, and probed its influence on behaviour. The results are overall in line with our main conclusions and particularly underscore the importance of neural tracking on response accuracy (see Supplementary Tables 14–17 for details).”

Minor comments:

1) You use the phrase “alternating regimes” in the significance statement – this could be read as implying that the tracking and alpha lateralization wax and wane in an alternating manner with “versus” each other – which is not true, right? They are independent. Maybe, rephrase for precision?

We thank the reviewer for this comment. We acknowledge that the use of this phrase may have been misleading. We have rephrased it accordingly:

[...]

“Closing the gap between hitherto separate lines of research, we used electroencephalography and a dual-talker task in a large sample of aging listeners to directly probe the functional relevance of state- and trait-level changes in these neural filter strategies to listening success. We demonstrate the co-existence of largely independent neural filters that establish complementary neurobiological implementation of selective attention.”

2) “synoptic”? Is that what you mean to say? As in a “summary” look?? Doesn’t seem to be the right word to me. Maybe “synergistic”?

This has been changed. Thank you.

3) I was a bit unclear on lines 448-450 – did you mean specifically elevated alpha with reduced tracking? Or vice versa? Or it can go both ways?

We thank the reviewer for this comment. See below for the updated, more precise description of the observed interactive effect of neural measures on response speed. To further aid interpretation, we added a visualization of the interaction to the supplements, printed below for the reviewer’s convenience.

As can be inferred from the interaction plot, the model did in fact predict the fastest responses for trials in which either relatively increased neural tracking co-occurred with relatively decreased alpha lateralization, or vice versa:

“Under selective but not divided attention, response speed depended on a combination of trial-to-trial variation in both alpha lateralization and neural tracking (LMM; spatial cue x ALI (within-subject effect) x neural tracking (within-subject effect): $\beta=0.08$, $SE=.03$, $p=.01$; see Supplementary Table 2). In short, responses were fastest in trials where relatively elevated levels in either neural tracking or alpha lateralization were paired with relatively reduced levels in the respective other neural measure thus highlighting the impact of two independent complementary filter solutions (see also Supplementary Fig. 5).”

Supplementary Fig. 5. Under selective attention, the combination of within-subject variability of neural tracking and alpha power lateralization during final word presentation predicts response speed. Predicted effect of alpha power lateralization on response shown for three different levels of neural tracking across all N=155 participants. Colored lines show group-level fixed effects, shaded colored areas indicate 95% confidence intervals. β : slope parameter estimate from general linear mixed-effects models.

4) Using time lags from -100 to 500 ms for the neural tracking is definitely sub optimal. You might have gotten improved sensitivity if you had used something like 50 – 250 ms (i.e., where there is actual signal).

We appreciate the reviewer’s comment on the potential impact of the chosen time lags on our neural tracking results. To address the concern that a wider range of time lags may have negatively impacted the sensitivity of our neural tracking measures, we ran an additional control analysis reconstructing envelopes based on the time lags of 50–250 ms that capture the most pronounced deflections of the estimated response function.

Using a linear mixed effect model, we compared the resultant neural tracking index values based on this shorter and the originally used longer time window. Importantly, neural tracking did not systematically vary as a function of the chosen time lags, presumably because the index measure reflects relative difference in reconstruction accuracy. So even if a wider range of time lags had led to

overall decreased reconstruction accuracies, it most likely would have affected the reconstruction of the attended and ignored envelope in a similar fashion.

For full transparency, we reference the results of this control analysis in the control analysis section of the main manuscript and have added the figure printed below to the supplements.

Please refer to our response to Major comment #6 above for a quote from the updated control analysis section.

Supplementary Fig 9. Comparison of single-subject (N=155) neural tracking index values using reconstruction accuracy (Pearson's r) based on different time lags. β : slope parameter estimate from general linear mixed-effects models. Left panel compared mean neural tracking per subject based on 50–250 ms vs. –100–500 ms. Right panel shows mean reconstruction accuracy of the attended envelope (top), and the ignored envelope (bottom) for both ranges of time lags. Error bars indicate 95 % confidence interval.

Typos:

Subtitle on line 411 – “power” not “lower”

This has been changed accordingly. Thank you.

Reviewer #2 (Remarks to the Author):

This study addresses neural measures of attentive listening to speech, in a large group of adults who span a wide range of age (39-80 years) and typical hearing loss. In recent years, two neural correlates of attending to speech in noise, commonly measured with EEG or MEG, have become increasingly recognized: modulation of alpha power lateralization between hemispheres and modulation of low-frequency speech envelope tracking. However, despite ample evidence of their involvement during attentive listening, there is little evidence relating these neural measures to each other or to their joint functional consequences behaviorally. The present study tackles this important, outstanding mechanistic question in the field. The authors take a remarkably comprehensive approach, incorporating how the nature of the attentional cue (spatial vs semantic) affects the neural metrics, and how the neural metrics relate to one another, to which ear is doing the listening, to behavior, and to other demographic variables (age, hearing loss). Furthermore, the study does all this in a well balanced manner, both across subjects and across individual trials, and over time within trials. It's beautifully, thoughtfully designed. Given all the experimental factors, the results are complex, but they basically demonstrate that spatially (but not semantically) informative attentional cueing increases both alpha lateralization and speech tracking in presumed auditory cortices. Importantly, alpha lateralization is largely uncorrelated with speech tracking strength (particularly on an individual trial basis, and regardless of spatial attention), and only speech tracking helps explain improved comprehension. Alpha lateralization and speech tracking thus appear to reflect somewhat independent attentional mechanisms, with alpha showing no behavioral impact in this task – a somewhat surprising result that holds across age and hearing status.

As noted above, this study addresses a deep, fundamental question in the field, and it is likely broadly relevant to other modalities too (e.g. vision). Methodologically it is well crafted, well executed, and utterly meticulous: the cohort is very large and well characterized, stimuli and tasks are cleverly designed, analyses directly address the hypotheses, good control analyses are run, and statistics are thorough and appropriate. The data and analysis code will evidently be posted upon publication, which will be a tremendous boon to the field and will support replication efforts.

We thank the reviewer for their enthusiastic review and for acknowledging the importance of the addressed research questions.

The main challenge is that this study raises more questions than it answers, and the reader is still left wondering about the original motivation: as the authors put it in the discussion, “how do the two neural filters relate to one another, and how do they influence listening success in a demanding real-life listening situation?” On this point, the reader would benefit from a clearer take-home message, even if it’s somewhat speculative.

We thank the reviewer for this helpful comment on how to better connect the current set of results with the research questions posed. In addressing this and the following related comment, we have substantially revised the discussion section to provide a clearer interpretation of the neural results and their differential effects on behaviour.

Please refer to our responses to the more specific comments below for more details.

MAJOR

It is unclear how to interpret the rich results. Examples include: If alpha lateralization and speech tracking are “independent” mechanisms, what distinct roles could they be serving? Other investigators have observed a relation between alpha lateralization and performance; why don’t we see it here?

In response to this very important comment, we have substantially revised those parts of the discussion that relate to the functional independence of alpha lateralization and neural speech tracking, as well as to their differential impact on behaviour.

In essence, we now emphasize more strongly that alpha lateralization and neural speech tracking instantiate neural filter solutions that are based on different mechanistic principles: the inhibition or controlled suppression of irrelevant input (i.e., alpha) versus the selective amplification of behaviourally relevant sensory information (i.e., neural speech tracking). The fact that we found these two mechanisms to be functionally independent may in fact be beneficial for the implementation of auditory attention as neural filter states may be more dynamically combined.

Below we quote the most relevant parts of the revised discussion for the reviewer’s convenience:

[...]

“There is preliminary evidence, mostly from between-subjects analyses, suggesting that the two neural filter strategies may exhibit a systematic relationship (Henry, Herrmann, Kunke, & Obleser, 2017; Kerlin, Shahin, & Miller, 2010; Lakatos et al., 2016; Tune et al., 2018; Wöstmann & Obleser, 2016). How the two neural filter strategies may be connected mechanistically is thus still an open question. We here asked whether concurrent changes in the two neural measures would imply a neural hierarchy in which alpha-driven controlled inhibition modulates the amplification of behaviourally relevant sensory information via selective neural tracking (Chapeton, Haque, Wittig, Inati, & Zaghoul, 2019; Spaak, Bonnefond, Maier, Leopold, & Jensen, 2012; van Kerkoerle et al., 2014).

[...]

The results from our most fine-grained single-trial level of analysis thus speak against a consistent relationship of momentary neural filter states. We observed instead the coexistence of two complementary but seemingly independent neurobiological solutions to the implementation of auditory selective attention.

How can this finding be reconciled with findings from previous electrophysiological studies (Henry et al., 2017; Kerlin et al., 2010; Lakatos et al., 2016) pointing towards a functional trade-off between neurobiological attentional filter mechanisms? And what could be an advantage to independent neural solutions for selective auditory attention?

[...]

Our between-subjects cross-correlation analysis appears to provide at least tentative support for a systematic relationship, in which peaks in neural tracking precede those in alpha lateralization. A closer inspection of the group-level temporal modulation of neural measures throughout sentence presentation, however, reveals some important differences to previous results. Whereas those studies reported an acyclic waxing and waning of neural entrainment and alpha power in response to rhythmic auditory stimulation (Lakatos et al., 2016; Tune et al., 2018; Wöstmann et al., 2016), in the current study, the two neural measures show different temporal dynamics: neural tracking gradually increases leading up to the final word, while alpha lateralization peaks at sentence and final word onset. The temporal dynamics of alpha lateralization, in particular, may point to strategic intermittent engagement of spatial attention in line with task demands (Bonacci et al., 2020).

Do these differences in temporal dynamics of the two neural filters challenge the existence of a systematic single-trial brain–brain relationship? Yes, but they also point to a potential benefit of independent neural filter solutions. If the two neural measures of auditory attention were indeed functionally unconnected as suggested by the current results, they would allow for a wider range of neural filter state configurations to flexibly adapt to the current task demands and behavioural goals. The co-existence of two independent but complementary filter mechanisms operating either via the selective amplification of relevant or via the controlled inhibition of irrelevant sounds, enables different modes of auditory attention to serve a listener’s goal in the face of complex real-life listening situations (Herrmann, Henry, Haegens, & Obleser, 2016; Schroeder et al., 2010; Schroeder & Lakatos, 2009)."

With respect to the absence of a link of alpha lateralization to behaviour, we now provide a more in-depth discussion of why previous studies have reported mixed evidence on this matter. We review evidence underscoring the multifaceted nature of human alpha oscillations that makes it particularly challenging to establish a robust brain-behaviour relationship:

[...]

"Using a sophisticated linear-model approach that probed the impact of both state- and trait-level modulation of neural filters on behaviour, we only found evidence for a direct influence of neural speech tracking but not alpha lateralization on behavioural performance even though all three measures were robustly modulated by the presence of a spatial cue (see Fig. 6). What could be the reason for the differential impact of neural measures on behaviour?

[...]

Despite a vast number of studies investigating the role of (lateralized) alpha oscillations in attentional tasks, the circumstances under which their top-down modulation may affect the behavioural outcome are still insufficiently understood (Gundlach, Moratti, Forschack, & Müller, 2020). Rather, the presence of a stable brain–behaviour relationship hinges on several factors.

[...]

In summary, it is still a matter of debate by which mechanistic pathway, and at which processing stage, the modulation of alpha power will impact behaviour. While it is (often implicitly) assumed that alpha oscillations impact behaviour via a modulation of neural excitability and thus early sensory processing, there is little evidence that shows a direct influence of alpha oscillation on changes in neural excitability and on subsequent behaviour (Gundlach et al., 2020; Iemi et al., 2021).

Lastly, the increase in alpha lateralization around final word presentation could at least partially reflect post-perceptual processes associated with response selection rather than the perceptual analysis itself (Kloosterman et al., 2019). The observed combined influence of neural tracking and alpha lateralization on response speed but not accuracy would seem compatible with such an interpretation (but see also (Hauswald, Keitel, Chen, Rösch, & Weisz, 2020) for the combined influence of non-lateralized alpha power and speech tracking on intelligibility in a non-spatial listening task).

Taken together, our results underscore the impact of prioritized sensory encoding of relevant sounds via selective speech tracking on listening performance and highlight the difficulty in establishing a comparable link for a neural signature as multifaceted as alpha oscillations (Clayton, Yeung, & Cohen Kadosh, 2018; Sadaghiani & Kleinschmidt, 2016; Womelsdorf, Valiante, Sahin, Miller, & Tiesinga, 2014)."

Why is semantically informed attention so ineffective in eliciting a neural effect, even while improving behavior – doesn't that call into question the directness of the causal relationship between tracking and comprehension? As I mention above, the reader could use much more help contextualizing these results within the original motivation.

We thank the reviewer for this comment that was overall in line with major concern #1 raised by Reviewer 1. For our full reply, we hope it is not too inconvenient when we refer the reviewer to our joint response to these related aspects above on p. 2f. of this letter.

Most directly pertaining to your concern, we there argue that a specific semantic cue probably had a stronger impact on the analysis of response alternative in the decision process than on the processing of the sentences themselves. This would also account for the absence of a semantic effect on the two peri-trial neural measures central to our analyses.

MINOR

1) I was confused when alpha and speech tracking are termed neural or auditory "filtering strategies". Elsewhere you use clearer terms such as "neurobiological measures of auditory attention" or "instantiation of attention at the neurobiological level".

We concur with the reviewer that any scientific report on the neural (or cognitive) mechanisms underlying a construct as complex as 'attention' should strive to be as precise as possible in its used terminology. Here, however, we would like to kindly disagree with the reviewer on the usefulness of the terms 'filter' or 'filter strategies' as we see them as important algorithmic implementations (or, as you rightly state, 'instantiations') to characterize the mechanistic details of how selective attention may be brought about at the neurobiological level. Notably, also, one of the seminal papers on auditory attention, Colin Cherry's original Cocktail party study (Cherry, 1953) explicitly referred to a 'filter' (p. 976) for solving this problem, which is the tradition we intended to allude to with the chosen terminology.

To pre-empt any misunderstanding, however, in the revised introduction section we now more clearly define our use of the term 'neural filter' for the current manuscript and emphasize more strongly that there are multiple nonexclusive neurobiological solutions to the instantiation of an attentional filter. For a quote of the updated introduction section, please refer to our response to major comment #4 by Reviewer 1 (p. 5 of this letter).

Prominently placed when first mentioning the neural measures in the Results section, we also now explicitly make use of the terminology the reviewer suggests:

"Using generalized linear mixed-effects models on single-trial data, we focus on two key neurobiological instantiations of auditory attention ..."

2) You claim that since alpha lateralization and tracking vary independently, this demonstrates a “functional trade-off”. Even with the grand average temporal relationship (tracking preceding alpha), that’s rather hard to prove given the stereotyped trial structure.

We agree with the reviewer that the use of this phrase particularly in the abstract or significance statement may have suggested a more systematic relationship of lateralized alpha power and neural speech tracking than is supported by the current results.

In the revised manuscript, we have rephrased the respective sentences together with our efforts to address the reviewer’s major comments stated above.

The relevant part of the abstract was revised as follows:

[...]

“First, we observed preserved attentional–cue-driven modulation of both neural filters across chronological age and hearing levels. Second, neural filter states varied independently of one another, demonstrating complementary neurobiological solutions of spatial selective attention. Stronger neural speech tracking but not alpha lateralization boosted trial-to-trial behavioural performance. Our results highlight the translational potential of neural speech tracking as an individualized neural marker of adaptive listening behaviour.”

3) The alpha that shows no relationship to tracking happens to peak at the end of the final word, which suggests that it could have as much to do with the subsequent remembering and responding rather than comprehending. This fits with the fact that final-word alpha did relate (jointly with neural tracking) to response speed but not accuracy. Kindly comment on this possibility.

We agree with the reviewer that this appears, at first glance, to be a plausible explanation given the temporal dynamic of alpha lateralization around and following final word onset, as well as its interactive effect on response speed.

At second glance, we think that there are several points that speak against such an interpretation: First, the probe period was well separated in time from the analysed time window of cue and sentence presentation, making a leakage of response-related neural signal into the analysed response less likely. Moreover, if the peak in lateralized alpha power was indeed mostly reflective of response preparation one would expect its peak to last until the actual response was given. However, as shown by Figure 3a, alpha lateralization actually decreases after the presentation of the last word but prior to the presentation of the probe screen. Lastly, our analysis focused on changes in alpha power within auditory cortex (i.e., at the source level), which renders it unlikely that we primarily captured neural signatures related to the response-preparatory motor processes.

We acknowledge however that the observed peak in lateralized alpha power could in principle be partially contaminated by post-perceptual processes linked to response selection rather than selective attention to final word.

In the revised manuscript, we happily discuss this possibility in the subsection on the links from neural measures to behaviour:

[...]

“In summary, it is still a matter of debate by which mechanistic pathway, and at which processing stage the modulation of alpha power, presumably reflecting a change in neural excitability, will impact behaviour (Iemi et al., 2021).

[...]

Lastly, the increase in alpha lateralization around final word presentation could at least partially reflect post-perceptual processes associated with response selection rather than the perceptual analysis itself (Kloosterman et al., 2019). The observed combined influence of neural tracking and alpha lateralization on response speed but not accuracy would seem compatible with such an interpretation (but see also (Hauswald et al., 2020) for the combined influence of non-lateralized alpha power and speech tracking on intelligibility in a non-spatial listening task).“

Reviewer #3 (Remarks to the Author):

The manuscript by Tune et al represents a behemoth of a study examining both neural tracking and alpha oscillations in aging subjects in a challenging listening task. The manuscript suggests that attention-driven neural tracking and alpha modulation are independent processes not affected by age.

The manuscript is well written, easily to understand and the results are transparent as evidenced by the supplement data. I find no issues with the statistical methods used by the authors – they are very appropriate for this line of research.

We thank the reviewer for his encouraging assessment of our study and for providing feedback on how to increase transparency of our methodological approach. Please see our detailed responses below.

A major finding was that neural tracking and alpha were independent is surprising. A “nice/simple” story would be that alpha primes the auditory cortex to be more sensitive to neural tracking .. this doesn’t appear to be the case. Although the authors mention Hauswald et al.’s paper that a joint model with alpha and speech tracking better predicted behaviour than either alone, some elaboration on this discrepancy would be beneficial.

We thank the reviewer for this comment as it relates to a point we did not emphasize strongly enough in the original manuscript: the very fact that we found the two neural measures to be statistically independent does not preclude their independent or even interactive effects on behaviour.

To the contrary, it is their statistical independence that allowed us to probe their joint effects in the first place. With respect to the study by Hauswald et al. this also means that a combined effect of alpha power and speech tracking on intelligibility as they have observed does not require a systematic relationship of the two neural measures.

Nevertheless, it still leaves the question of why Hauswald et al. observed a combined effect of alpha power and neural speech tracking on intelligibility whereas we only observed an effect of neural speech tracking on accuracy (and response speed).

As there are several differences with respect to the experimental design and cognitive processes under investigation, as well as the statistical analysis, we can only speculate which factors may have had the biggest impact. As Hauswald and colleagues investigated the effect of different degradation levels on speech intelligibility in a non-spatial setting, they modelled the influence of alpha power itself rather than its lateralization across hemispheres. Moreover, their design was more suited to tap into the concept of listening effort rather than selective spatial attention. Finally, in their ‘combined’ model they did not test for the additive or interactive effect of alpha power and neural speech tracking on behaviour as we did, but predicted speech intelligibility from a ratio of the two neural measures.

While an in-depth account on these differences is beyond the scope of the discussion, please see the quote in response to the third comment of Reviewer 2 in which we also express more clearly that Hauswald et al. examined the role of (non-lateralized) alpha power and neural tracking in a non-spatial listening task. (see p. 11 of this letter).

I find very little room for any criticism. Perhaps only minor clarification:

The use of single trial neural tracking has not been well explored in the past and the use of this method for alpha single trial correlations is insightful. How were the “sliding windows” implemented? i.e., shape, overlap...

Thank you for bringing the need for a more in-depth description to our attention. The respective section has been revised as follows:

“Since individual sentences differed in length, for visualization and statistical analysis, we mapped their resulting neural tracking time courses onto a common time axis expressed in relative (percent) increments between the start and end of a given stimulus. We first assigned each sample to one of 100 bins covering the length of the original sentence in 1% increments. We then averaged across neighbouring bins using a centred rectangular 3% sliding window (1% overlap). The same procedure was applied to the time course of alpha power lateralization following up-sampling to 125 Hz.”

What were the range of lambdas (in the TRF script) that were finally used in the neural tracking measure? Was the same lambda used for everyone or was it optimized per subject?

We thank the reviewer for highlighting the lack of clarity in the description of our TRF training procedure. Please see below for the revised paragraph of the method section. In essence, to avoid overfitting, we added the regularization term λmI to the time-lagged neural response matrix. This term involved a subject-specific constant m , calculated as the across-trial mean trace of the $R^T R$ with R being the time-lagged version of the neural response matrix r . The mean trace m proved relatively stable across subjects which in turn meant that the optimal ridge parameter differed only very little between subjects.

“We used all parcels within the bilateral auditory ROI and time lags τ in the range of -100 ms to 500 ms to compute envelope reconstruction models using ridge regression (Hoerl & Kennard, 1970):

$$g = (R^T R + \lambda m I)^{-1} R^T s \quad (3)$$

where R is a matrix containing the sample-wise time-lagged replication of the neural response matrix r , λ is the ridge parameter for regularization, I is the identity matrix, and m is a subject-specific scalar representing the mean of the trace of $R^T R$ (Biesmans, Das, Francart, & Bertrand, 2017); (Fiedler et al., 2017). The same grid of ridge parameters ($\lambda = 10^{-5}, 10^{-4}, \dots, 10^{10}$) was used across subject, and m proved to be relatively stable across subjects (387.2 ± 0.18 , mean \pm sd). The optimal ridge value of $\lambda = 1$ was determined based on the average Pearson’s correlation coefficient and mean squared error of the reconstructed and actually presented envelope across all trials and subjects.”

**The use of auditory cortex ROI for the alpha was a bit surprising given that a lot of previous work found dominant generator in the parietal regions for auditory spatial processing. However, the use of a parietal control ROI puts my mind at ease. Were the responses at parietal larger/more robust than auditory cortex?
Andrew Dimitrijevic**

We kindly refer the reviewer to Supplementary Fig. 8 on alpha power lateralization in the inferior parietal control region included in the supplemental information and printed below along with Figure 3a for the reviewer’s convenience.

Supplementary Fig 8. Attentional modulation of 8–12 Hz inferior parietal alpha power throughout the trial for all N=155 participants. Brain models indicate the spatial extent of the inferior parietal control region of interest (shown in green). Purple traces show the alpha lateralization index (ALI) for the informative (solid dark purple line) and uninformative spatial cue (dashed light purple line), each collapsed across semantic cue levels. Positive values indicate relatively higher alpha power in the hemisphere ipsilateral to the attended/-probed sentence compared to the contralateral hemisphere. Shaded grey area shows the time window of sentence presentation.

Figure 3. Informative spatial cue elicits increased alpha power lateralization before and during speech presentation.

(a) Attentional modulation of 8–12 Hz auditory alpha power throughout the trial for all N=155 participants. Brain models indicate the spatial extent of the auditory region of interest (shown in red). Purple traces show the alpha lateralization index (ALI) for the informative (solid dark purple line) and uninformative spatial cue (dashed light purple line), each collapsed across semantic cue levels. Positive values indicate relatively higher alpha power in the hemisphere ipsilateral to the attended/-probed sentence compared to the contralateral hemisphere. Shaded grey area shows the time window of sentence presentation.

In the absence of statistical testing, visual comparison of neural responses across the two regions of interest actually suggests more pronounced increases in lateralized alpha power, particularly relative to spatial-cue onset and final word presentation, in the auditory ROI.

REVIEWER COMMENTS

Reviewer #1 (Remarks to the Author):

The authors have done an excellent job of responding to my comments on the previous version of their manuscript (which I already thought was a very good manuscript). I have no further concerns on this nice work.

Reviewer #2 (Remarks to the Author):

I thank the authors for their thoughtful and thorough responses. This report is a fantastic contribution to this important topic - in many ways a model for others. I have no further comments.

-Lee M Miller

Reviewer #3 (Remarks to the Author):

All of my comments have been sufficiently addressed